# RULE-BOTTLENECK RL: LEARNING TO DECIDE AND EXPLAIN FOR SEQUENTIAL RESOURCE ALLOCATION VIA LLM AGENTS

## ABSTRACT

Deep Reinforcement Learning (RL) has demonstrated remarkable success in solving sequential resource allocation problems, but often suffers from limited explainability and adaptability—barriers to integration with human decision-makers. In contrast, LLM agents, powered by large language models (LLMs), provide human-understandable reasoning but may struggle with effective sequential decision making. To bridge this gap, we introduce Rule-Bottleneck RL (RBRL), **the first LLM agent framework for resource allocation problems that jointly optimizes language-based decision policy and explainability.** At each step within RBRL, an LLM first generates candidate rules—language statements capturing decision priorities tailored to the current state. RL then optimizes rule selection to maximize environmental rewards and explainability, with the LLM acting as a judge. Finally, the LLM chooses the action (optimal allocation) based on the rule. We provide conditions for RBRL performance guarantees as well as the finite-horizon evaluation gap of the learned RBRL policy. Furthermore, we provide evaluations in real-world scenarios, particularly in public health, showing that RBRL not only improves the performance of baseline LLM agents, but also approximates the performance of Deep RL while producing more desirable human-readable explanations. We conduct a human survey validating the improvement in the quality of the explanations.

## 1 INTRODUCTION

Sequential resource allocation is a fundamental problem in many domains, including healthcare, finance, public policy, and operations research (Considine et al., 2025; Boehmer et al., 2024; Yu et al., 2024; Balaji et al., 2019). This task involves allocating limited resources over time while accounting for dynamic changes and competing demands. Deep reinforcement learning (RL) is an effective method to optimize decision-making in resource allocation offering scalable high-reward policies (Yu et al., 2021; Talaat, 2022; Xiong et al., 2023), albeit generally providing action recommendations without human-readable reasoning and explanations. Such lack of interpretability poses a major challenge in critical high-stake domains where decisions must be transparent, justifiable, and in line with human decision-makers to ensure trust and compliance with ethical and regulatory standards.

For example, in healthcare settings, doctors may need to decide whether to prioritize intervention for Patient A or Patient B based on their current vital signs (Boehmer et al., 2024). An RL algorithm might suggest: *"Intervene with Patient A "* with the implicit goal of maximizing the value function. However, the underlying reasoning may not be clear to the doctors, leaving them uncertain about the factors influencing the decision (Milani et al., 2024). For doctors, a more effective suggestion could be risk-based with specific information, e.g., *"Patient A's vital signs are likely to deteriorate leading to higher potential risk compared to Patient B, so intervention with Patient A is prioritized"* (Gebrael et al., 2023; Boatin et al., 2021).

LLM agents (Sumers et al., 2024), on the other hand, leverage large language models (LLMs) for multi-step decision-making using reasoning techniques like chain of thought (CoT) (Wei et al., 2022). They enable natural language goal specification (Du et al., 2023) and enhance human understanding (Hu & Sadigh, 2023; Srivastava et al., 2024). However, agents based solely on LLM reasoning often

Figure 1: Overview of the framework of RBRL for joint sequential decision-making and explanation generation at time instance $t$. Starting with current state $\mathbf{s}_t$, a state-to-language descriptor generates $\texttt{lang}(\mathbf{s}_t)$, which is used to create the input prompt $\mathbf{p}_t$. The LLM processes $\mathbf{p}_t$ to produce a thought $\boldsymbol{\tau}_t$ and a set of candidate rules $\mathcal{R}_t$. An attention-based policy network selects a rule $\mathbf{a}_t^{\text{rule}}$ obeying the budget constraint $B(\mathbf{s}_t)$, which is used by LLM to derive an executable action $\mathbf{a}_t^{\text{env}}$ for the environment and a human-readable explanation $\boldsymbol{\ell}_t^{\text{expl}}$, while also providing a rule reward $r_t^{\text{rule}}$. The environment transitions to the next state $\mathbf{s}_{t+1}$, returning an environment reward $r_t^{\text{env}}$. This process is repeated iteratively at subsequent time steps.

struggle with complex sequential decision-making out of the box (Furuta et al., 2024), making RL a crucial tool for grounding to specific tasks (Carta et al., 2023; Tan et al., 2024; Wen et al., 2024; Zhai et al., 2024).

Consequently, aiming to combine the strengths of both deep RL and LLM agents, we pose the following question:

> *Can we design an LLM agent framework that can simultaneously perform sequential resource allocation and provide human-readable explanations?*

Similar to the celebrated index policy for prioritizing arms in resource allocation problems (Whittle, 1988), we explore the potential of using rules-based prioritization in resource allocation tasks. In the context of LLM agents, rules are defined as "structured statements" that capture prioritization among choices in a given state, aligning with the agent's goals (Srivastava et al., 2024). Building on this, we propose a novel LLM agent framework called Rule-Bottleneck Reinforcement Learning (RBRL), which integrates the strengths of LLM and RL to bridge the gap between decision-making and interpretability. RBRL provides an agent framework (as shown in Figure 1) that *simultaneously* makes sequential resource allocation decisions and provides human-readable explanations, in contrast to prior work that generates post-hoc explanations for a learned policy (Peng et al., 2022; Milani et al., 2024). RBRL leverages LLMs to generate candidate rules and employs RL to optimize policy, allowing the creation of effective decision policies while simultaneously providing human-understandable explanations. RBRL aims to increase efficiency and avoid the computational cost of directly fine-tuning LLM agents, which can be highly challenging in interactive environments due to the heavy computational costs and the complexity of token-level optimization (Rashid et al., 2024).

Our contributions are summarized as follows. *First*, LLMs are leveraged to generate a diverse set of rules according to the environment state, where each rule serves as a prioritization strategy for individuals in resource allocation, enhancing interpretability in decision-making. *Second*, we extend the conventional environmental state-action space by integrating the rules into states generated by LLMs, creating a novel framework that enables RL to operate on a richer, more interpretable decision structure. *Third*, we introduce an attention-based training framework that maps states and rules to queries and keys of a cross-attention network. The rule selection process is optimized by a policy network trained using the Soft Actor-Critic (SAC) algorithm (Haarnoja et al., 2018), ensuring robust and efficient decision-making. In particular, the LLM also acts as a feedback mechanism, providing guidance during RL exploration to improve policy optimization and promote more effective learning. To the best of our knowledge, this is the first work to jointly optimize decision-making and explanation generation in constrained RL tasks.

We evaluate our method in environments from three real-world domains: HeatAlerts, where resources are allocated to mitigate extreme heat events; WearableDeviceAssignment, for distributing monitoring devices to patients; and BinPacking, which models allocating limited space in containers under constraints to optimize space utilization and minimize overflows. Using cost-effective LLMs such as gpt-4o-mini (OpenAI, 2024) and Llama 3.1 8B (Meta AI, 2024), we first assess decision performance by comparing RBRL with pure RL methods and language agent baselines.

| Step 1: Generate Thoughts |
|---|
| Two example thoughts:
- There are only four warnings remaining in the budget.
- The current heat index is high, and issuing alert could raise public awareness. |
| **Step 2: Generate Rules Based on Thoughts and the Current State** |
| An example rule:
- Background : Maintaining a balance in warning issuance is crucial for future effectiveness
- Rule: If there are 3 or more warnings remaining, issue a warning when the heat index is above 105 F.
- State Relevance: There are 4 warnings remaining, allowing for proactive issuance given the current heat index of 107 F. |

(a) Examples of generated rules for the Heat Alert Issuance task.

| Example Language Wrapper for Heat Alert Issuance |
|---|
| *Task:* Assist policymakers in deciding when to issue public warnings to protect against heatwaves. Your goal is to minimize the long-term impact on health and mortality. Your decision should be based on the remaining budget, weather conditions, day of the week, past warning history, and remaining warnings for the season. The goal is to issue warnings when they are most effective, minimizing warning fatigue and optimizing for limited resources. |
| *Action:* A single integer value representing the decision: 1 = issue a warning, 0 = do not issue a warning. Warning can only be issued if the 'Remaining number of warnings/budget' is positive. Response in JSON format. For example: {'action': 1}. |
| *State:* Remaining warning budget: 4, - Current date and day of summer: 2008-07-10, - Current heat index: 107 F. |

(b) Examples of language wrapper, containing task description, available actions and current state.

Figure 2: Examples of task prompts and generated rules for `HeatAlerts` domain.

We then evaluate explanation quality through a human survey conducted under IRB approval. The results demonstrate `RBRL`'s effectiveness in both decision quality and interpretability.

## 2 RELATED WORK

Our work is positioned at the intersection of RL for resource allocation, LLM agents, and Explainable RL (XRL). While traditional RL methods effectively optimize rewards for resource allocation (Boehmer et al., 2024), they often lack the interpretability required for high-stakes domains. Conversely, LLM agents that provide reasoning (Wei et al., 2022) can struggle with sequential optimization. Our framework is novel compared to hierarchical approaches that use LLMs for high-level planning (Carta et al., 2023; Szot et al., 2023), as RBRL is the first to treat the natural-language rule as a primary output, jointly optimizing for both decision-making performance and the rule's quality as an explanation. Furthermore, unlike post-hoc or attribution-based XRL methods that analyze decisions after the fact (Guo et al., 2021; Chen et al., 2024), RBRL provides intrinsic explanations, as the rule is a functional component within the decision-making loop. A detailed discussion of related literature is provided in Appendix B.

## 3 PRELIMINARY, KEY CONCEPTS, AND PROBLEM FORMULATION

### 3.1 PRELIMINARY: RESOURCE-CONSTRAINED ALLOCATION

Resource-constrained allocation tasks are usually formulated as a special type of constrained Markov Decision Process, which is defined by the tuple $\langle \mathcal{S}, \mathcal{A}, P, R, C, h, \gamma \rangle$, where $\mathcal{S}$ denotes a state space and $\mathcal{A}$ denotes a finite action space. The transition probability function, specifying the probability of transitioning to state $\mathbf{s}' \in \mathbb{R}^{d_1}$ after taking action $\mathbf{a} \in \mathbb{R}^{d_2}$ in state $\mathbf{s}$, is $P(\mathbf{s}'|\mathbf{s}, \mathbf{a}) : \mathcal{S} \times \mathcal{A} \times \mathcal{S} \rightarrow \Delta(\mathcal{S})$, $R(\mathbf{s}, \mathbf{a}) : \mathcal{S} \times \mathcal{A} \rightarrow \mathbb{R}$ represents the reward function, defining the immediate reward received after taking action $\mathbf{a}$ in state $\mathbf{s}$, and we let $C(\mathbf{s}, \mathbf{a}) : \mathcal{S} \times \mathcal{A} \rightarrow \mathbb{R}^{d_3}$ be the immediate cost incurred after taking action $\mathbf{a}$ in state $\mathbf{s}$. Often, each dimension $i \in [d_2]$ in $\mathbf{a}$ is either 0 or 1 in resource-constrained allocation tasks. In addition, $h$ is the time horizon and $\gamma \in [0, 1]$ denotes the discount factor, which determines the present value of future rewards.

The goal is to find a policy $\pi : \mathcal{S} \rightarrow \Delta(\mathcal{A})$ that maximizes the expected cumulative discounted reward while satisfying the cost constraints with a budget function $B : \mathcal{S} \rightarrow \mathbb{R}^{d_3}$:

$$\pi^* = \arg\max_{\pi} \mathbb{E}_{\pi} J(\pi) := \left[ \sum_{t=1}^{h} \gamma^{t-1} R(\mathbf{s}_t, \mathbf{a}_t) \right], \ s.t. \ \forall t \in [h] : C(\mathbf{s}_t, \mathbf{a}_t) \leq B(\mathbf{s}_t). \quad (1)$$

## 3.2 KEY CONCEPTS FOR RULE-BASED LLM AGENTS

Our challenge is to design a rule-based LLM agent that jointly optimizes a language policy to both solve the optimization problem and improve explanation quality—a direction rarely explored. We next introduce the key concepts and terminologies underlying our main contribution.

**LLM Agent**  For our LLM agent, the action space includes internal language actions $\tilde{\mathcal{A}} = \mathcal{A} \cup \mathcal{L}$ (Yao et al., 2023). The LLM agent has two types of internal language actions: First, *thoughts* $\mathbf{a}^{\text{thought}} \in \mathcal{L}$, are reasoning traces from the current problem state used to inform environment action selection $\mathbf{a}^{\text{env}} \in \mathcal{A}$. Second, *explanations* $\boldsymbol{\ell}^{\text{expl}}$, are generated from actions and thoughts to enhance human trust and interpretability (Zhang et al., 2023), a focus of this work.

**Rules**  Thoughts are useful to highlight relevant aspects of a problem. However, they often lack detailed information to identify the next optimal action. In this work, we will consider "rules" $\mathbf{a}^{\text{rule}} \in \mathcal{L}$, which are structured language statements derived from thoughts that generally take the form "[if/when][do/prioritize] ". More formally, each rule $\mathbf{a}^{\text{rule}}$ consists of a triple (`background`, `rule_statement`, `state_relevance`). Figure 2a shows examples of generated rules from one of the domains used in the experiments.

**Task and Constraints Description**  Language agents require: (1) a language description of the environment and the agent's goal, denoted `task`, containing the available actions for the task; (2) a function describing the state of the environment in natural language, denoted `lang` $: \mathcal{S} \to \mathcal{L}$. At each state $\mathbf{s}_t$, these descriptors are used to construct a natural language prompt $\mathbf{p}_t = f(\texttt{task}, \texttt{lang}(\mathbf{s}_t))$. Figure 2b exemplifies language wrapper generated for one of the environments in our experiments.

**Rule-based Language Policy**  The objective is to jointly optimize the reward and explainability of the environment. Hence, we have an LLM agent-driven policy $\pi_{\text{LLM}}$ for online interaction with the environment:

$$\mathbf{a}_t^{\text{thought}} \sim \pi_{\text{LLM}}(\mathbf{a}_t^{\text{thought}} \mid \mathbf{p}_t), \ \ \mathbf{a}_t^{\text{rule}} \sim \pi_{\text{LLM}}(\mathbf{a}_t^{\text{rule}} \mid \mathbf{a}_t^{\text{thought}}, \mathbf{p}_t),$$
$$\mathbf{a}_t^{\text{env}} \sim \pi_{\text{LLM}}(\mathbf{a}_t^{\text{env}} \mid \mathbf{a}_t^{\text{rule}}, \mathbf{a}_t^{\text{thought}}, \mathbf{p}_t), \ \ \boldsymbol{\ell}_t^{\text{expl}} \sim \pi_{\text{LLM}}(\boldsymbol{\ell}_t^{\text{expl}} \mid \mathbf{a}^{\text{env}}, \mathbf{a}_t^{\text{rule}}, \mathbf{p}_t). \tag{2}$$

The rule acts as a "bottleneck" to the action and explanation. In the next section, we will introduce RBRL, which allows an RL-based learnable selection policy $\pi_\theta$ choosing among a set of dynamically generated candidate rules.

## 3.3 PROBLEM STATEMENT

We aim to increase the quality of $\boldsymbol{\ell}^{\text{expl}}$ while also optimizing decision-making by selecting rules that encourage both good quality explanations and high reward. To achieve this goal, we construct a surrogate explainabilty "rule reward" $R_{\text{LLM}}^{\text{rule}}(\mathbf{a}^{\text{rule}})$ using an LLM as judge (Shen et al., 2024; Bhattacharjee et al., 2024; Gu et al., 2024), which will be detailed in Section 4. Then, we propose the following augmented optimization objective under the joint environment/rule reward as $\tilde{R}(\mathbf{s}_t, \mathbf{a}_t^{\text{env}}) = R(\mathbf{s}_t, \mathbf{a}_t^{\text{env}}) + R_{\text{LLM}}^{\text{rule}}(\mathbf{a}^{\text{rule}})$:

$$\max_\pi \mathbb{E}_\pi \tilde{J}(\pi) := \left[ \sum_{t=1}^h \gamma^{t-1} \tilde{r}_t \right], \ s.t. \text{ constraint in (1)}, \tag{3}$$

where $\tilde{r}_t = \tilde{R}(\mathbf{s}_t, \mathbf{a}_t^{\text{env}})$. We emphasize that LLMs cannot fully replace the ultimate human assessment, but they they provide a scalable alternative during the optimization process.

## 4 RULE-BOTTLENECK REINFORCEMENT LEARNING (RBRL)

In this section, we propose RBRL, a novel LLM agent based on the key concepts in Section 3.2, which leverages the strengths of LLMs and RL to achieve both interpretability and robust sequential decision-making for (3), thereby achieving our goal of jointly optimizing policies and explanations for resource-constrained allocation in (1).

**Algorithm Overview**  The framework of RBRL shown in Algorithm 1 involves four steps: (1) RULE SET GENERATION (line 3), where the LLM processes the state-task $\mathbf{p}_t$ to create candidate rules $\mathcal{R}_t$

---

**Algorithm 1** RBRL

---

**Require:** Rule-selection policy $\pi_\theta$; and Replay buffer $\mathcal{B}$.
 1: **Initialization:** Initial state $\mathbf{s}_0$ and task-state prompt $\mathbf{p}_0$.
 2: **for** $t = 0, \ldots, \texttt{max\_iters} - 1$ **do**
 3:     Generate rule candidates $\mathcal{R}_t$ using CoT from $\mathbf{p}_t$ and $\mathbf{a}_t^{\text{thought}}$. // Section 4.1
 4:     Select rule $\mathbf{a}_t^{\text{rule}}$ using the RL policy $\pi_\theta$ from $\mathcal{R}_t$ and $\mathbf{s}_t$. // Section 4.2
 5:     Generate the environment action $\mathbf{a}_t^{\text{env}}$ with the LLM from $\mathbf{a}_t^{\text{rule}}$, $\mathbf{p}_t$, and previous thoughts.
 6:     Apply the action in the environment and obtain new state $\mathbf{s}_{t+1}$ and environment reward $r_t^{\text{env}}$.
 7:     Generate explanation with the LLM from $\mathbf{a}_t^{\text{env}}$, $\mathbf{r}_t^{\text{rule}}$, $\mathbf{p}_t$, and previous thoughts.
 8:     Generate rule reward $r_t^{\text{rule}}$ with the LLM as judge. // Section 4.3
 9:     Update the prompt $\mathbf{p}_{t+1}$ from $\mathbf{s}_{t+1}$, and the constraints $C$ and budget $B$.
10:     Append transition to the replay buffer $\mathcal{B} \leftarrow \mathcal{B} \cup \{(\tilde{\mathbf{s}}_t, \mathbf{a}_t^{\text{rule}}, \tilde{r}_t, \tilde{\mathbf{s}}_{t+1})\}$.
11:     Sample from the replay buffer and update the policy network $\pi_\theta(\mathbf{a}_t^{\text{rule}}|\tilde{\mathbf{s}}_t)$. // Section 4.4
12: **end for**

---

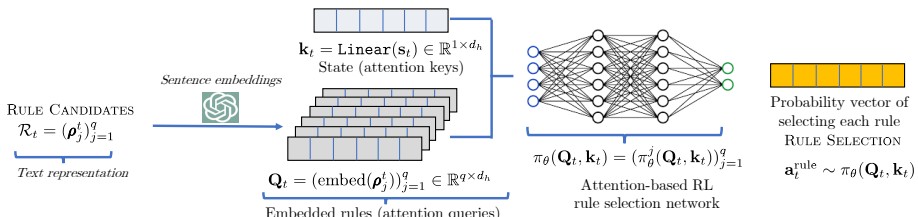

Figure 3: Overview of the RULE SELECTION step. The current state is encoded as a key vector, while candidate rules are encoded as Queries using a text embedding API (e.b., BERT sentence embedding). An attention-based policy network $\pi_\theta$ computes a probability distribution over the candidate rules, enabling the selection of the most suitable rule for decision-making and explanation.

for action selection; (2) RULE SELECTION (line 4), where an attention-based RL policy $\pi_\theta$ selects the best rule $\mathbf{a}_t^{\text{rule}} \in \mathcal{R}$; (3) DECISION, RULE REWARD AND EXPLANATION (lines 5-8), where the LLM generates an environment action $\mathbf{a}_t^{\text{env}}$ and based on the chosen rule $\mathbf{a}_t^{\text{rule}}$ gives a human-readable explanation $\boldsymbol{\ell}_t^{\text{expl}}$; (4) REINFORCEMENT LEARNING (line 11), where it updates the policy $\pi_\theta$ based on collected data with standard RL algorithm Haarnoja et al. (2018) and the combined environment and rule reward $\tilde{r}_t$. Algorithm 1 details the entire process. Further sections elaborate on these steps.

## 4.1 RULE SET GENERATION

The rule generation process seeks to create interpretable and actionable guidelines for decision-making. Under this framework, a set of candidate rules $\mathcal{R}_t$ is generated according to $\mathcal{R}_t \sim \pi_{\text{LLM}}(\mathcal{R}_t|\mathbf{p}_t, \mathbf{a}_t^{\text{thought}})$. To enhance interpretability, each rule is accompanied by a rationale explaining the reasoning behind the decision. The LLM is instructed to generate rules as a JSON format, which is common for integration of LLMs with downstream applications (Shen et al., 2024). An example generated rule is given in Figure 2a. See Figure 12 in the Appendix for the prompt templates used rules generation.

## 4.2 RULE SELECTION

In this step, rules are converted from text to vector form, and a trainable attention-based policy network $\pi_\theta$ provides the probability distribution for sampling a rule. Figure 3 illustrates the process, with a detailed procedure in Algorithm 2 of the Appendix. Below are the major components of the architecture of $\pi_\theta$. We propose to base the architecture on cross-attention layers (Bahdanau et al., 2015; Vaswani et al., 2017), with the state acting as the keys and values, and the rules as the queries. This allows to learn from the embedding representations of rules, and efficiently handle dynamically changing number of rules if needed.

**State Representation** The numeric state is projected by a linear layer: $\mathbf{k}_t = \texttt{Linear}(\mathbf{s}_t) \in \mathbb{R}^{1 \times d_h}$, with $d_h$ being to denote the architecture hidden dimension.

**Rule Candidate Embedding** Each rule in the list of rule candidates $\mathcal{R}_t = \{\boldsymbol{\rho}_1^t, \boldsymbol{\rho}_2^t, \ldots, \boldsymbol{\rho}_q^t\}$ is embedded into a numeric representation using a Sentence Embedding language model (e.g., SentenceBERT (Reimers & Gurevych, 2019)) and further processed by a projection layer similar to the state representation. This results in a *query* matrix $\mathbf{Q}_t \in \mathbb{R}^{q \times d_h}$.

**Attention-based Policy Network $\boldsymbol{\pi_\theta}$** The vector $\mathbf{k}_t$, serving as keys, engages with the rule embeddings $\mathbf{Q}_t$, acting as queries, via a cross-attention mechanism to derive a hidden state representation $\mathbf{h}_t^{(1)} = \texttt{Attention}(\mathbf{Q}_t, \mathbf{k}_t^\top, \mathbf{k}_t^\top) \in \mathbb{R}^{q \times d_h}$, computed as $\texttt{Attention}(\mathbf{Q}_t, \mathbf{k}_t^\top, \mathbf{k}_t^\top) =$ softmax $\left( \frac{\mathbf{Q}_t \mathbf{k}_t^\top}{\sqrt{d_h}} \right) \mathbf{k}_t^\top$, which facilitates the rule candidate vector embeddings in attending to the environment state. Subsequently, we sequentially apply self-attention layers to the hidden representation $\mathbf{h}^{(k+1)} = \texttt{Attention}(\mathbf{h}_t^{(k)}, \mathbf{h}_t^{(k)}, \mathbf{h}_t^{(k)})$, enabling the rule embeddings to attend to one another to rank an optimal candidate. Ultimately, following $K - 1$ self-attention layers, a final linear layer converts the rule representations into logit vectors $\boldsymbol{\alpha}_\theta^t = \texttt{Linear}(\mathbf{h}_t^{(k)}) \in \mathbb{R}^q$ used for the computation of the probability of selecting each rule.

**Rule Selection** The policy distribution over the rules is calculated as: $\pi_{\theta,i}(\mathbf{Q}_t, \mathbf{k}_t) = \frac{\exp(\alpha_{\theta,i}^t(\mathbf{Q}_t, \mathbf{k}_t))}{\sum_{j=1}^q \exp(\alpha_{\theta,j}^t(\mathbf{Q}_t, \mathbf{k}_t))}$, $i = 1, \ldots, q$. Therefore, a rule is selected at random from the distribution: $\mathbf{a}_t^{\text{rule}} \sim \texttt{Categorical}(\mathcal{R}; (\pi_{\theta,i}(\mathbf{Q}_t, \mathbf{k}_t))_{i=1}^q)$.

### 4.3 DECISION, RULE REWARD, AND EXPLANATION

Upon selection of rule $\mathbf{a}_t^{\text{rule}}$, the LLM determines the action to be applied within the environment $\mathbf{a}_t^{\text{env}} \sim \pi_{\text{LLM}}(\mathbf{a}_t^{\text{env}}|\mathbf{a}_t^{\text{rule}}, \mathbf{a}_t^{\text{thought}}, \mathbf{p}_t)$, ensuring concordance with the chosen strategy. Subsequently, the LLM formulates an explanation $\boldsymbol{\ell}_t^{\text{expl}} \sim \pi_{\text{LLM}}(\boldsymbol{\ell}_t^{\text{expl}}|\mathbf{a}^{\text{env}}, \mathbf{a}^{\text{rule}}, \mathbf{a}^{\text{thought}}, \mathbf{p}_t)$ contingent upon the rule. Figure 12 in the Appendix illustrates the prompt template employed to generate both the action and explanation.

This procedure concurrently produces the rule reward $R_{\text{LLM}}^{\text{rule}}(r_t^{\text{rule}})$, used for RL in the next step. This rewards is derived from using the LLM as a judge to answer the following three questions: $\texttt{ER}_1$. Without providing $\mathbf{a}_t^{\text{env}}$, is $\mathbf{a}_t^{\text{rule}}$ sufficient to predict the optimal action? $\texttt{ER}_2$. Does $\mathbf{a}_t^{\text{rule}}$ contain enough details about the applicability of the rule to current state? $\texttt{ER}_3$. Given $\mathbf{a}_t^{\text{env}}$, is $\mathbf{a}_t^{\text{rule}}$ compatible with this selection? Each question scores as 0 if negative or 1 if positive. The rule reward is calculated as $r_t^{\text{rule}} = R_{\text{LLM}}^{\text{rule}}(\mathbf{a}_t^{\text{rule}}) \propto (1/3) \sum_i \texttt{ER}_i$. Refer to Figure 12 in the Appendix for the full prompt.

### 4.4 POLICY UPDATE THROUGH RL

**Augmented state space** Traditional RL frameworks fail to directly return a policy based on current environment state due to intermediate steps: generating the rule set $\mathcal{R}_t$, mapping rules $\mathbf{a}_t^{\text{rule}}$ to actions $\mathbf{a}_t^{\text{env}}$ in an LLM-driven environment. RBRL addresses this issue by creating an augmented state $\tilde{\mathbf{s}}_t := (\mathbf{s}_t, \mathcal{R}_t)$ with transition dynamics $P(\tilde{\mathbf{s}}_{t+1}|\tilde{\mathbf{s}}_t, \mathbf{a}_t^{\text{rule}})$, integrating rules into the state space for reasoning over both the environment's dynamics and decision rules $\mathbf{a}_t^{\text{rule}}$. The following theorem explains the transition computation.

**Theorem 4.1.** *The state transition of the* RBRL *MDP can be calculated as*

$$P(\tilde{\mathbf{s}}_{t+1}|\tilde{\mathbf{s}}_t, \mathbf{a}_t^{rule}) = P(\mathcal{R}_{t+1}|\mathbf{s}_{t+1}) \times \int_{\mathbf{a}} P(\mathbf{s}_{t+1}|\mathbf{a}^{env}, \mathbf{s}_t) \cdot P(\mathbf{a}^{env}|\mathbf{a}_t^{rule}, \mathbf{s}_t) d\mathbf{a}^{env}, \qquad (4)$$

*where* $P(\mathcal{R}_{t+1}|\mathbf{s}_{t+1}) = \pi_{LLM}(\mathcal{R}_{t+1}|\mathbf{p}_t, \boldsymbol{\tau}_t)$ *is the probability of the LLM generating rule set* $\mathcal{R}_{t+1}$ *provided the state* $\mathbf{s}_{t+1}$, $P(\mathbf{s}_{t+1}|\mathbf{a}^{env}, \mathbf{s}_t)$ *is the original environment dynamics, and* $P(\mathbf{a}^{env}|\mathbf{a}_t^{rule}, \mathbf{s}_t) = \pi_{LLM}(\mathbf{a}^{env}|\mathbf{p}_t, \mathbf{a}_t^{rule})$ *is the probability of the LLM selecting the environment action* $\mathbf{a}^{env}$.

**Policy update step** The attention-based policy network in Section 4.2 is optimized using the standard SAC algorithm, which balances reward maximization with exploration. The policy network in SAC is updated by minimizing the KL divergence between the policy and the Boltzmann distribution induced by Q networks $Q_{\phi_i}, \forall i = 1, 2$, which is expressed as

$$L_\pi(\theta) = \mathbb{E}_\mathcal{D} \left[ \beta \log \pi_\theta(\mathbf{a}_t^{\text{rule}}|\tilde{\mathbf{s}}_t) - \min_{i=1,2} Q_{\phi_i}(\tilde{\mathbf{s}}_t, \mathbf{a}_t^{\text{rule}}) \right], \qquad (5)$$

where $\beta$ is a temperature parameter. The detailed implementation for SAC update procedure is detailed in Algorithm 3 in Appendix D.

## 5 PERFORMANCE GUARANTEE

In this section, we derive and prove conditions under which RBRL can learn the optimal task policy, as well as characterize the potential trade-off between explainability and task performance when rewarding rules for higher explainability.

**Proposition 5.1** (Rule Set Coverage). *Let $\mathcal{A}$ be a finite action space and $Q^*(\mathbf{s}, \mathbf{a}^{env})$ the optimal state-action value function, with $\mathbf{a}^{env,*}(\mathbf{s}) := \arg\max_{\mathbf{a}^{env} \in \mathcal{A}} Q^*(\mathbf{s}, \mathbf{a}^{env})$ denoting the optimal action at state $\mathbf{s}$. Given state $\mathbf{s}$, an LLM samples $N$ rules independently from a conditional distribution $\pi_{LLM}(\cdot \mid \mathbf{s})$, and each rule $\boldsymbol{\rho}_i$ maps $\mathbf{s}$ to an action $\mathbf{a}_i^{env} \sim \pi_{LLM}(\mathbf{a}_i^{env} | \boldsymbol{\rho}_i, \mathbf{s})$. Assume there exists $\delta > 0$ and $\eta_s \in (0,1]$ such that: $\mathbb{P}_{\boldsymbol{\rho}_i \sim \pi_{LLM}(\cdot | \mathbf{s})}[Q^*(\mathbf{s}, \mathbf{a}_i^{env}) \geq Q^*(\mathbf{s}, \mathbf{a}^{env,*}(\mathbf{s})) - \delta] \geq \eta_s$. Define the $\delta$-optimal rule set as: $\mathcal{R}^\delta(\mathbf{s}) := \{\boldsymbol{\rho}_i : Q^*(\mathbf{s}, \mathbf{a}_i^{env}) \geq Q^*(\mathbf{s}, \mathbf{a}^{env,*}(\mathbf{s})) - \delta\}$. Then with high probability over the sampled rules, there at least has a rule $\boldsymbol{\rho}_i$ and the induced action $\mathbf{a}_i^{env} \sim \pi_{LLM}(\mathbf{a}_i^{env} | \boldsymbol{\rho}_i, \mathbf{s})$ that satisfies:*

$$\mathbb{E}[Q^*(\mathbf{s}, \mathbf{a}^{env,*}(\mathbf{s})) - Q^*(\mathbf{s}, \mathbf{a}_i^{env})] \leq \delta + \epsilon_{worst} \cdot (1 - \eta_s)^N, \quad (6)$$

*where $\epsilon_{worst} := \max_{\boldsymbol{\rho} \notin \mathcal{R}^\delta(\mathbf{s})} (Q^*(\mathbf{s}, \mathbf{a}^{env,*}(\mathbf{s})) - Q^*(\mathbf{s}, \mathbf{a}_i^{env}))$ is the worst-case loss outside the $\delta$-optimal set.*

*Remark 5.2.* Proposition 5.1 states the *rule diversity* property in the rule candidate set such that the best possible action (when $\delta \to 0$) is included is guaranteed with high probability when number of rules $N$ goes large. This is crucial in guaranteeing that RBRL can learn a near-optimal policy with high probability (with optimality when $\delta = 0$ and $\eta_s = 1$). See Section E in Appendix for more detail. We also numerically evaluate rule-coverage in Figure 8 of Appendix G.5.

Define the T-step value function $V_{\mathcal{M}'}^{\pi,T}(\mathbf{s}_0) = [\sum_{t=0}^{T-1} \gamma^t R_t^{\mathcal{M}'}(\mathbf{s}_t, \pi(\mathbf{s}_t)) | \mathbf{s}_0]$, where $R^{\mathcal{M}'}$ is the reward function in $\mathcal{M}'$. We will denote the original MDP as $\mathcal{M}$ and use $\tilde{\mathcal{M}}$ to denote the MDP for the RBRL agent with transition function as in Theorem 4.1 and reward $\tilde{R}$. We have the following theorem.

**Theorem 5.3.** *The evaluation gap $Gap(T, s_0) := V_{\mathcal{M}}^{\pi^*, T}(\mathbf{s}_0) - V_{\mathcal{M}}^{\pi_{RBRL}, T}(\mathbf{s}_0)$ of RBRL is bounded as*

$$Gap(T, s_0) = V_{\mathcal{M}}^{\pi^*, T}(\mathbf{s}_0) - V_{\tilde{\mathcal{M}}}^{\pi_{RBRL}, T}(\mathbf{s}_0) + V_{\tilde{\mathcal{M}}}^{\pi_{RBRL}, T}(\mathbf{s}_0) - V_{\mathcal{M}}^{\pi_{RBRL}, T}(\mathbf{s}_0) \leq \lambda \cdot \frac{1 - \gamma^T}{1 - \gamma}, \quad (7)$$

*where $\lambda$ is a constant depending on the magnitude of the rule reward, and, with a slight notational abuse, $V_{\mathcal{M}}^{\pi_{RBRL}, T}$ is the value of the RBRL policy when seen as a policy in the original MDP mapping states to actions (i.e., by integrating out the rule generation and action selection via LLMs.)*

*Remark 5.4.* This analysis focuses on the evaluation gap between the optimal policy $\pi^*$ under the original MDP $\mathcal{M}$ and the policy $\pi_{RBRL}$, captures the suboptimality of using $\pi_{RBRL}$ instead of the true optimal policy $\pi^*$, assuming RBRL is optimized under the extended MDP $\tilde{\mathcal{M}}$ (with same transitions as $\mathcal{M}$ but additional rule-based reward). It can be decomposed into two interpretable terms. The first part captures the optimism of using $\pi_{RBRL}$ under the extended MDP $\hat{\mathcal{M}}$ rather than the original MDP, which is non-positive. The second part quantifies the accumulated reward difference induced by the additional explanation rewards when using the same RBRL policy in both MDPs.

## 6 EXPERIMENTS & HUMAN SURVEY

In this section, we evaluate RBRL and empirically show that it can achieve a joint improvement in both reward and explainability over comparable baselines. We briefly summarize these environments here, with additional details in Appendix F.1.

**Domains** We evaluate RBRL in three main distinct resource-constrained allocation domains:

▷ WearableDeviceAssignment: We use two environments, Uganda and MimicIII, from the vital sign monitoring domain introduced by Boehmer et al. (2024), modeling the allocation of limited wireless devices among postpartum mothers as an MDP setting. ▷ HeatAlerts: We use the

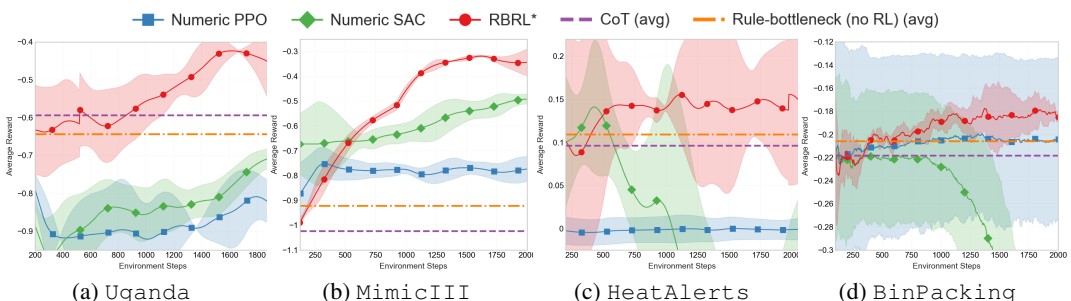

(a) `Uganda`   (b) `MimicIII`   (c) `HeatAlerts`   (d) `BinPacking`

Figure 4: Results from Q1 using ChatGPT 4o-mini. The plots show the mean and standard error across three seeds, using exponentially weighted moving averages ($\lambda_{\text{ema}} = 200$).

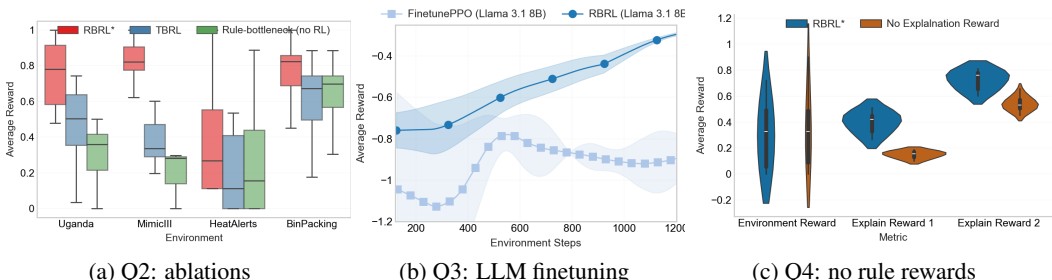

(a) Q2: ablations   (b) Q3: LLM finetuning   (c) Q4: no rule rewards

Figure 5: Additional experiments and ablations. (a) Comparison of `RBRL` with thoughts-based RL (TBRL) and the baseline rule-based LLM without RL training; (b) comparison against LLM finetuning with PPO at the token level from the environment reward with CoT generation for the `Mimic`; (c) shows the effect of removing the rule reward in the `HeatAlerts` environments. For (a) and (c), we show distribution of rewards in the last 20% training steps.

`weather2alert` environment from Considine et al. (2025), which formulates issuing heat alerts as a constrained MDP to reduce hospitalization risk from the alert. ▷ `BinPacking` We adopt the online stochastic `BinPacking`: environment introduced by Balaji et al. (2019), which Sequentially places arriving items into bins with fixed capacity to minimize total waste, following the online stochastic formulation. Detailed domain description can be found in Appendix F.1.

## 6.1 ENVIRONMENT REWARD OPTIMIZATION

We discuss the main results and refer to Appendix F for the detailed experiment setup and Appendix G for additional experiments. Unless otherwise specified, we use `gpt-4o-mini` as LLM due to its reasonable cost and high performance.

*Q1. Did `RBRL` optimize the reward function?* `RBRL` is compared to CoT (Wei et al., 2022) for language reasoning and PPO (Schulman et al., 2017) for numeric states. Figure 4 indicates `RBRL` outperforms CoT, showing RL-optimized rule selection improves decision-making. `RBRL` also exceeds PPO in all environments with equal environment steps, suggesting a better online learning performance. Notice that `RBRL` is compatible with a baseline LLM trained for advanced reasoning techniques (e..g, GRPO Shao et al. (2024)). However, GRPO or similar cannot be used directly in MDPs. Nevertheless, our experiments with the comparable GPT o3 (see Appendix G) prove that RBRL can also help improve reasoning models in our tasks.

*Q2. Did structured rules help optimization?* We conduct two ablation studies on structured rules. First, we benchmark the use of structured rules without RL, called baseline `Rule-bottleneck(no RL)`, which is shown in Equation (2)-(5). Next, we compare `RBRL` with a variant optimizing unstructured thoughts, termed thoughts-based RL (TBRL). The implementation mimics `RBRL`, utilizing a candidate pool $\mathcal{P}$ with the CoT prompt. Results in Figure 5a show that comparing `RBRL` with `RulesLLMOnly` highlights RL training gains, suggesting rule generation alone does not explain `RBRL`'s performance. Additionally, significant improvements over `TBRL` suggest optimizing structured rules is more effective than optimizing free reasoning traces.

*Q3. How does* `RBRL` *compare to token-level LLM finetuning with RL?* We implement LLM finetuning on a Llama 3.1 8B model, termed `FinetunePPO`. A value head is trained on final hidden states, with KL divergence from a reference model as regularization (Ziegler et al., 2019). CoT is generated, followed by an action query, optimizing both. Training runs for 18 hours on 3 seeds using an A100 40G GPU (1200 steps/seed). For fair comparison, `RBRL` is also run on Llama 3.1 8B. Figure 5b shows `RBRL` outperforms the flatter trend of finetuning, indicating better online learning. Moreover, `RBRL` runs on a regular laptop, whereas `FinetunePPO` requires specialized hardware and takes 4× longer per step. Due to compute limits, results are shown only for the less noisy `MimicII` domain.

**Additional comparison with XRL benchmarks** We further compare `RBRL` against a representative XRL method that also targets joint optimization and intrinsic interpretability: Decision Diffusion Trees (DDTs) (Silva et al., 2020). As shown in Table 1, `RBRL` is consistently competitive and often outperforms the tree-based baseline across most domains, particularly in the early stages of training, underscoring its sample efficiency. Although DDT achieves a higher average reward than `RBRL` in `HeatAlerts`, it exhibits substantially higher variance, highlighting the greater stability of `RBRL`.

## 6.2 HUMAN SURVEY AND EXPLAINABILITY

*Q4. Did* `RBRL` *increase the explainability of explanations?* A survey with 40 participants was conducted to assess explanation quality, detailed in Appendix J. Each prompt included the task, state, and action space as originally given to the LLM, followed by actions and explanations from the CoT agent and the `RBRL` agent, with-

Table 1: XRL Baselines Results Table

| Dataset (@steps) | RBRL | SAC | PPO | DDT | DDT w/rules |
|---|---|---|---|---|---|
| Uganda (@500) | **−0.56 ± 0.18** | −0.83 ± 0.14 | −0.91 ± 0.14 | −1.01 ± 0.20 | −1.20 ± 0.31 |
| Uganda (@2500) | **−0.60 ± 0.20** | −0.75 ± 0.14 | −0.74 ± 0.05 | −1.28 ± 0.35 | −1.20 ± 0.30 |
| MimicIII (@500) | **−0.36 ± 0.05** | −0.61 ± 0.11 | −0.78 ± 0.05 | −0.92 ± 0.10 | −1.02 ± 0.10 |
| MimicIII (@2500) | **−0.39 ± 0.07** | −0.43 ± 0.10 | −0.64 ± 0.10 | −0.97 ± 0.11 | −0.99 ± 0.13 |
| HeatAlerts (@500) | 0.14 ± 0.11 | −0.04 ± 0.33 | 0.00 ± 0.01 | **0.22 ± 0.25** | 0.15 ± 0.29 |
| HeatAlerts (@2500) | 0.13 ± 0.14 | 0.05 ± 0.04 | 0.00 ± 0.01 | **0.38 ± 0.57** | 0.38 ± 0.56 |
| BinPacking (@500) | **−0.03 ± 0.00** | −0.03 ± 0.00 | −0.03 ± 0.00 | −0.19 ± 0.03 | −0.19 ± 0.04 |
| BinPacking (@2500) | **−0.03 ± 0.00** | −0.06 ± 0.00 | −0.03 ± 0.00 | −0.21 ± 0.02 | −0.21 ± 0.02 |

out disclosing agent types. Participants were asked to choose preference for explanation A, B, or none. Prompts were split between `WearableDeviceAssignment` and `HeatAlerts` domains. Figure 6 shows results, favoring `RBRL`'s explanations in both domains, with a detailed breakdown in J. An additional experiment with an LLM judge using a large `gpt-4o` model showed strong agreement with humans, preferring `RBRL`'s explanations in all cases.

**Discussion on Explainability** The trustworthiness of explanations is a core challenge in XAI. Following recent work (Kunz & Kuhlmann, 2024; Parcalabescu & Frank, 2023; Jacovi & Goldberg, 2020), we highlight three concepts: *Plausibility:* whether an explanation is convincing to humans (validated via our survey, Figure 6). *Consistency:* whether the stated reason logically entails the action (Appendix G.3). *Faithfulness:* whether the explanation reflects the true decision mechanism.

Our work is motivated by the gap between plausibility and faithfulness in post-hoc methods. By design, `RBRL` ensures consistency: explanations factually follow the State → Rule → Action pipeline, where the rule is the verifiable cause of the action. As shown in Table 8 in Appendix G.3, including reasoning traces does not affect the decision, confirming that consistency stems from the state and rule rather than LLM self-explanation. While our experiments validate consistency, establishing faithfulness—verifying the LLM's internal reasoning for rule generation—remains an open challenge.

*Q5. What was the effect of the rule reward?* During training of `RBRL`, rules received rewards from two prompts. We examine an ablation without this reward. Figure 5c illustrates results for the `HeatAlerts` environment, noted for high variance and a challenging reward function. We extended training to 5k steps to understand these dynamics. Without rule reward, environment reward remains steady (slightly increasing), but explainability scores drop significantly. Re-

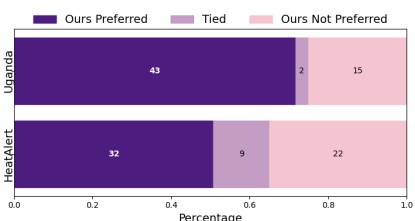

Figure 6: Results from the human survey.

fer to Section 4.3 for the definition of the rule reward metrics. A decline in metric 1 indicates that rules are less predictive of the optimal actions. A decline in metric 2 suggests rules lack detailed applicability to the current problem state, indicating more generic rather than specialized rule selection. Metric 3 (not shown) was always 1 in all steps, indicating the limitations of directly evaluating post hoc explanations. Although judged by the LLM, these results are encouraging, as our previous experiment showed alignment between the LLM and human assessments.

## ETHICS AND REPRODUCIBILITY STATEMENTS

The authors of this work adhere to the ICLR Code of Ethics. Our research involves domains with significant ethical considerations, particularly in healthcare and public policy, which we have carefully addressed. Our work includes a human survey to evaluate the quality of explanations, which was conducted under Institutional Review Board (IRB) approval.

We have taken extensive measures to ensure the reproducibility of our research. The complete source code for our framework, including environment implementations and experiment scripts, is provided as supplementary material, with an anonymous link included at the beginning of the Appendix. All algorithmic details and hyperparameters for our proposed method (RBRL) and all baselines are comprehensively documented in Appendix D and Appendix F.

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
