# Supplementary Materials

In this supplementary material, we provide additional details omitted from the main paper for brevity. We also present extended experimental results to further support our findings. The complete source code for this work is available at: `https://anonymous.4open.science/r/rule-bottleneck-reinforcement-learning-6F0D`.

## A    LLM USAGE DISCLOSURE

We used LLMs solely as algorithmic components within our proposed method. No LLMs were employed for writing, editing, or generating the content of this manuscript.

## B    RELATED WORK

Our work intersects with three distinct areas within the RL literature. We discuss related work in each of these domains.

**RL for Sequential Resource Allocation**    RL has been widely studied for constrained resource allocation across domains. In maternal health, Boehmer et al. (2024) apply RL to a restless multi-armed bandit (RMAB) problem (Whittle, 1988) to compute stochastic intervention probabilities. Also in an RMAB setting, Xiong et al. (2022) propose a model-based RL approach that prioritizes users via an index and allocates resources under budget constraints. In public health, Considine et al. (2025) propose RL to optimize extreme heat warnings under a budget on the number of possible alerts. Other works include multi-agent RL for robotic warehouse allocation (Shen et al., 2023) and exogenous MDPs for cloud resource management (Sinclair et al., 2023). While these methods optimize rewards effectively, they often lack interpretability—critical for deployment in sensitive domains requiring trust, transparency, and accountability.

**RL and LLM Agents**    One stream of research in LLM agents (Sumers et al., 2024) has developed somewhat independently of RL, with works like ReAct prompting (Yao et al., 2023) extending chain-of-thought (CoT) (Wei et al., 2022) to action settings. These works have focused on tasks such as open-ended web navigation (Putta et al., 2024), social simulations (Park et al., 2023), and virtual assistants (Vezhnevets et al., 2023). Meanwhile, LLM agents have also been proposed for dealing with complex Markov decision processes such as GLAM (Carta et al., 2023), TWOSOME Tan et al. (2024), BAD (Wen et al., 2024), and AgentGym (Xi et al., 2024), which use LLM finetuning techniques in RL environments with a reward function. While our work is related to hierarchical methods that leverage LLMs for high-level planning (Wang et al., 2023; Szot et al., 2023), our framework is novel in its objective. Unlike prior work that uses language solely to guide a policy toward high task rewards, RBRL is the first to treat the language-based "rule" as a primary output, jointly optimizing for both decision-making performance and the rule's quality as a human-readable explanation via a dedicated reward signal.

**Explainable RL (XRL)**    Early XRL relied on methods like decision trees and concept-based explanations (Das et al., 2023), but these struggled with scalability in dynamic environments (Poeta et al., 2023). Recent advances introduced LLMS for post-hoc explanations, such as explaining decision paths from policy trees (Zhang et al., 2023) or adding language descriptions to RL policies (Colas et al., 2022). However, these approaches focus on interpreting pre-existing policies rather than enabling LLMs to generate inherently explainable decisions, with challenges in aligning explanations to human reasoning (Singh et al., 2024). By contrast, inherently (also known as intrinsically) interpretable policies are those that have internal representation that allow explanations (Peng et al., 2022; Milani et al., 2024). Our work sits within this literature by using LLM reasoning traces as the basis for environment action selection. Other methods like EDGE (Guo et al., 2021) and RICE (Cheng et al., 2024) are primarily attribution-based; they identify which inputs (e.g., pixels or state features) were most critical to a decision. Similarly, SelfIE (Chen et al., 2024) provides a post-hoc, mechanistic explanation of an LLM's internal mechanics (hidden states). In contrast, RBRL generates high-level, user-facing policy rules that are functional components within the RL loop, providing an explanation of the agent's intent.

## C   IMPACT STATEMENT AND LIMITATIONS

This work advances the development of transparent AI systems for high-stakes decision-making in domains like healthcare, public policy, industry, and many other applications. By enabling LLM agents to generate human-readable rules and explanations while attaining reward maximization via RL, RBRL improves trust and accountability, critical for ethical deployment in settings where lives and resources are at stake. While the framework prioritizes alignment with human reasoning, potential risks include over-reliance on imperfect LLM-generated rules or explanations that may inadvertently obscure biases in training data. Mitigation requires rigorous validation of rules by domain experts and ongoing monitoring of LLM outputs. Additionally, RBRL 's reliance on LLMs raises computational and accessibility challenges in resource-constrained environments. By addressing these considerations, this research contributes to safer, more equitable AI systems that empower—rather than replace—human decision-makers. To further validate the interpretability of our method, we obtained IRB approval and conducted a human subject study to evaluate the quality of the generated explanations.

Notice that the Uganda dataset used in this study is derived from a simulator that models vital sign trajectories of patients, as provided by Boehmer et al. (2024). Importantly, this simulator only replicates vital sign transitions and does not include any feature information or identifying details of real patients. Thus, the data generated by the simulator cannot be traced to or represent actual individuals, ensuring privacy and ethical compliance. We emphasize that this is purely a simulated patient study; and recognize that for any next steps towards real world use, there is a need to conduct rigorous simulation studies on a large scale with real patient data, with detailed assessments of potential biases, verification of policy convergence and its robustness to distribution shifts in patient populations, and making necessary adjustments. Beyond that, there will be a need to obtain ethics and regulatory approval to test the policy in a real-world setting for future comprehensive field testing, addressing issues of participant consent and privacy; and ultimately there would need to be sufficient human oversight for any future deployment.

**Interpretability vs. Performance Tradeoff**   Various works acknowledge the trade-off between interpretability and performance (Rudin, 2019). In practice, prioritizing interpretability is crucial in practice in many high-stake applications: an approach that we subscribe to in this work. For example, in the clinical AI domain, high-performing black-box systems often face rejection in clinical workflows due to distrust (Shevtsova et al., 2024; DuBois, 2019) as physicians require transparency to validate recommendations and uphold ethical accountability, as mandated by regulatory frameworks (e.g., (EPC, 2024; CSL, 2024)). Interpretable models enable clinicians to audit biases, adapt decision logic to local contexts, and iteratively refine recommendations-fostering collaborative decision-making over reliance on inflexible oracles-whereas opaque policies are prone to failure under real-world distribution shifts(Rudin, 2019; Doshi-Velez & Kim, 2017; Shevtsova et al., 2024; DuBois, 2019). Empirical surveys show clinicians favor models that enable shared decision-making, error accountability, and ethical oversight despite modest performance penalties (Shevtsova et al., 2024)-a critical stance in high-stakes healthcare environments where trust and adaptability outweigh narrow efficiency gains.

## D   ALGORITHMIC DETAILS

In this section, we present the detailed pseudocodes for Rule_Search in Algorithm 2 and the SAC for attention-based policy network in Algorithm 3.

Algorithm 2 outlines the process of rule selection using attention-based policies. First, each rule candidate $\boldsymbol{\rho}_i^t$ is embedded into a numeric vector $\mathbf{q}_i^t$ using a sentence embedding technique (e.g., Sentence-BERT), forming a query matrix $\mathbf{Q}_t$. The state $\mathbf{s}_t$ is also converted into a numeric vector $\mathbf{k}_t$. Cross-attention is applied between $\mathbf{Q}_t$ and $\mathbf{k}_t$ to generate an attention representation $\mathbf{h}$, which may optionally be refined using a self-attention mechanism. A linear layer processes to $\mathbf{h}$ produce score vector $\boldsymbol{\alpha}_\theta^t$. These scores define the policy distribution $\pi_\theta$, from which a rule $\mathbf{a}_t^{\text{rule}}$ is sampled. This attention-based approach ensures efficient selection of rules by leveraging contextual relationships between the state and rule candidates. For the implementation, the attention layer is realized using the multi-headed attention module from Vaswani et al. (2017). We incorporate a dropout layer, fixed

at 0.05 for the experiments, along with SiLU activation and layer normalization, which are excluded from the notation for brevity.

---

**Algorithm 2** `Rule_Search`: Rule Selection via Attention-Based Policies

---

**Require:** Numeric state representation $\mathbf{s}_t \in \mathbb{R}^{d_s}$ and rule set $\mathcal{R}_t = \{\boldsymbol{\rho}_i^t\}_{i=1}^q$. Hidden dimension $d_h$.
1: Embed each rule $\boldsymbol{\rho}_i^t \in \mathcal{R}_t$ into a numeric vector using sentence embeddings $\mathbf{q}_i^t = \texttt{embed}(\boldsymbol{\rho}_i^t) \in \mathbb{R}^{d_h}$ (e.g., Sentence-BERT) and stack to form a query matrix $\mathbf{Q}_t \in \mathbb{R}^{d_h \times q}$. // *Rule Candidate Embedding*
2: The state $\mathbf{s}_t$ is projected by a linear layer with SiLU activation: $\mathbf{k}_t = \texttt{SiLU}(\text{Linear}(\mathbf{s}_t)) \in \mathbb{R}^{1 \times d_h}$, with $d_h$ being to denote the architecture hidden dimension. // *State Representation*
3: Use cross-attention to obtain $\mathbf{h} = \text{CrossAttention}(\mathbf{Q}_t, \mathbf{k}_t, \mathbf{k}_t) \in \mathbb{R}^{q \times d_h}$.
4: (Optional) Further apply a self-attention network $\mathbf{h} \leftarrow \text{SelfAttention}(\mathbf{h})$.
5: Apply linear layer to obtain logits vector $\boldsymbol{\alpha}_\theta^t = \text{Linear}(\mathbf{h}) \in \mathbb{R}^{q \times 1}$.
6: Calculate the policy distribution: $\pi_{\theta,i}(\mathbf{Q}_t, \mathbf{k}_t) = \frac{\exp(\alpha_{\theta,i}^t(\mathbf{Q}_t, \mathbf{k}_t))}{\sum_{j=1}^q \exp(\alpha_{\theta,j}^t(\mathbf{Q}_t, \mathbf{k}_t))}$, where $\alpha_{\theta,i}^t$ is $i$-th element in $\boldsymbol{\alpha}_\theta^t$.
7: Sample rule $\mathbf{a}_t^{\text{rule}} \sim \texttt{Categorical}(\mathcal{R}; (\pi_{\theta,i}(\mathbf{Q}_t, \mathbf{k}_t))_{i=1}^q)$. // *Rule Selection with Attention*
8: **Return** $\mathbf{a}_t^{\text{rule}}$.

---

**Algorithm 3** SAC for Attention-based Policy Network

---

1: Initialize Q networks $Q_{\phi_1}, Q_{\phi_2}$ and policy network $\pi_\theta$ with random parameters $\phi_1, \phi_2, \theta$.
2: Initialize target Q networks $\bar{Q}_{\bar{\phi}_1}, \bar{Q}_{\bar{\phi}_2}$ with weights $\bar{\phi}_1 \leftarrow \phi_1, \bar{\phi}_2 \leftarrow \phi_2$.
3: Initialize temperature parameter $\beta$ and target entropy $\mathcal{H}_{\text{target}}$; Initialize replay buffer $\mathcal{D}$.
4: **for** episode $= 1, \ldots, M$ **do**
5:    Initialize environment and observe initial state $\tilde{\mathbf{s}}_1$.
6:    **for** step $t = 1, \ldots, T$ **do**
7:       Sample action $\mathbf{a}_t^{\text{rule}} \sim \pi_\theta(\cdot | \tilde{\mathbf{s}}_t)$.
8:       Execute action $\mathbf{a}_t^{\text{rule}}$, observe reward $\tilde{r}_t$ and next state $\tilde{\mathbf{s}}_{t+1}$.
9:       Store transition $(\tilde{\mathbf{s}}_t, \mathbf{a}_t^{\text{rule}}, \tilde{r}_t, \tilde{\mathbf{s}}_{t+1})$ in replay buffer $\mathcal{D}$.
10:      **if** enough samples in $\mathcal{D}$ **then**
11:        Sample a mini-batch of transitions $(\tilde{\mathbf{s}}_t, \mathbf{a}_t^{\text{rule}}, \tilde{r}_t, \tilde{\mathbf{s}}_{t+1})$ from $\mathcal{D}$.
12:        Compute target Q values: $y_t = \tilde{r}_t + \gamma \mathbb{E}_{\mathbf{a}_{t+1}^{\text{rule}} \sim \pi_\theta(\cdot | \tilde{\mathbf{s}}_{t+1})} \left[ \min_{j=1,2} \bar{Q}_{\bar{\phi}_j}(\tilde{\mathbf{s}}_{t+1}, \mathbf{a}_{t+1}^{\text{rule}}) - \alpha \log \pi_\theta(\mathbf{a}_{t+1}^{\text{rule}} | \tilde{\mathbf{s}}_{t+1}) \right]$.
13:        Update Q networks by minimizing:
14:        $L_Q(\phi_i) = \mathbb{E}_{(\tilde{\mathbf{s}}_t, \mathbf{a}_t^{\text{rule}}, \tilde{r}_t, \tilde{\mathbf{s}}_{t+1})} \left[ (Q_{\phi_i}(\tilde{\mathbf{s}}_t, \mathbf{a}_t^{\text{rule}}) - y_t)^2 \right]$ for $i = 1, 2$.
15:        Update policy network by minimizing:
16:        $L_\pi(\theta) = \mathbb{E}_{\tilde{\mathbf{s}}_t, \mathbf{a}_t^{\text{rule}} \sim \pi_\theta} \left[ \beta \log \pi_\theta(\mathbf{a}_t^{\text{rule}} | \tilde{\mathbf{s}}_t) - \min_{j=1,2} Q_{\phi_j}(\tilde{\mathbf{s}}_t, \mathbf{a}_t^{\text{rule}}) \right]$.
17:        Update temperature parameter by minimizing:
18:        $L_\beta(\beta) = \mathbb{E}_{\tilde{\mathbf{s}}_t, \mathbf{a}_t^{\text{rule}} \sim \pi_\theta} \left[ -\beta \left( \log \pi_\theta(\mathbf{a}_t^{\text{rule}} | \tilde{\mathbf{s}}_t) + \mathcal{H}_{\text{target}} \right) \right]$.
19:        Update target Q networks:
20:        $\bar{\phi}_i \leftarrow \tau \phi_i + (1 - \tau) \bar{\phi}_i$ for $i = 1, 2$.
21:      **end if**
22:    **end for**
23: **end for**

---

Algorithm 3 presents the SAC algorithm tailored for training an attention-based policy network in selecting the desired rule. This method combines entropy-regularized policy optimization with a structured approach to handle rule-selection effectively. The algorithm begins with the initialization of key components: Q networks $Q_{\phi_1}, Q_{\phi_2}$, target Q networks $\bar{Q}_{\phi_1}, \bar{Q}_{\phi_2}$, and a policy network $\pi_\theta$. Random parameters are assigned to these networks, and the target Q networks are synchronized with the initial Q networks. A temperature parameter $\alpha$ is initialized to regulate the entropy $\mathcal{H}_{target}$ in the policy objective, ensuring a balance between exploration and exploitation. A replay buffer $\mathcal{D}$ is set

up to store transition data. Notice the entropy is defined as

$$\mathcal{H}_{target} = -\sum_{i=1}^{q} \pi_\theta(\mathbf{a}_t^{\text{rule}}|\mathbf{k}_t, \mathbf{Q}_t) \log \pi_\theta(\mathbf{a}_t^{\text{rule}}|\mathbf{k}_t, \mathbf{Q}_t). \tag{8}$$

During training, each episode starts with the initialization of the environment, and the agent observes the initial state $\tilde{\mathbf{s}}_1$. At every time step, the policy network generates an action $\mathbf{a}_t^{\text{rule}}$ based on the current state. This action is executed in the environment, resulting in a reward $\tilde{r}_t$ and a state transition to $\tilde{\mathbf{s}}_{t+1}$. These transitions are stored in the replay buffer for optimization. When sufficient transitions are available in the buffer, the algorithm samples a mini-batch of transitions and computes the target Q values. The target Q values incorporate entropy regularization and are computed using the minimum of the target Q networks to ensure stability. The Q networks are updated by minimizing the mean squared error between the predicted Q values and the computed targets. The policy network is optimized by minimizing a loss function that combines the policy entropy with the expected Q value, ensuring a stochastic and exploratory policy. The temperature parameter $\beta$ is updated to maintain the desired balance between exploration and exploitation. Finally, the target Q networks are softly updated to stabilize training. This iterative process continues across episodes and time steps, progressively refining the policy network to achieve optimal rule selection.

# E  MATHEMATICAL DETAILS

## E.1  PROOF OF THEOREM 4.1

In this section, we provide the detailed proofs for Theorem 4.1. We start with the following equation

$$P(\mathcal{R}_{t+1}|\mathbf{s}_{t+1}) \cdot \int_a P(\mathbf{s}_{t+1}|\mathbf{a}^{\text{env}}, \mathbf{s}_t) \cdot P(\mathbf{a}^{\text{env}}|\mathbf{a}_t^{\text{rule}}, \mathbf{s}_t) d\mathbf{a}^{\text{env}}$$

$$= P(\mathcal{R}_{t+1}|\mathbf{s}_{t+1}) \cdot \underbrace{\int_a P(\mathbf{s}_{t+1}|\mathbf{a}^{\text{env}}, \mathbf{s}_t, \mathcal{R}_t, \mathbf{a}_t^{\text{rule}}) \cdot P(\mathbf{a}^{\text{env}}|\mathbf{a}_t^{\text{rule}}, \mathbf{s}_t) d\mathbf{a}^{\text{env}}}_{(a)\ P(\mathbf{s}_{t+1}|\mathbf{a}^{\text{env}}, \mathbf{s}_t, \mathcal{R}_t, \mathbf{a}_t^{\text{rule}}) = P(\mathbf{s}_{t+1}|\mathbf{a}^{\text{env}}, \mathbf{s}_t)}$$

$$= P(\mathcal{R}_{t+1}|\mathbf{s}_{t+1}) \cdot \underbrace{\int_a P(\mathbf{s}_{t+1}|\mathbf{a}^{\text{env}}, \mathbf{s}_t, \mathcal{R}_t, \mathbf{a}_t^{\text{rule}}) \cdot P(\mathbf{a}^{\text{env}}|\mathbf{s}_t, \mathcal{R}_t, \mathbf{a}_t^{\text{rule}}) d\mathbf{a}^{\text{env}}}_{(b)\ P(\mathbf{a}^{\text{env}}|\mathbf{s}_t, \mathcal{R}_t, \mathbf{a}_t^{\text{rule}}) = P(\mathbf{a}^{\text{env}}|\mathbf{s}_t, \mathbf{a}_t^{\text{rule}})}$$

$$= P(\mathcal{R}_{t+1}|\mathbf{s}_{t+1}) \cdot \underbrace{\int_a P(\mathbf{s}_{t+1}, \mathbf{a}^{\text{env}}|\mathbf{s}_t, \mathcal{R}_t, \mathbf{a}_t^{\text{rule}}) d\mathbf{a}^{\text{env}}}_{P(A|B,C) \cdot P(B|C) = P(A,B|C)}$$

$$= \underbrace{P(\mathcal{R}_{t+1}|\mathbf{s}_{t+1}) \cdot P(\mathbf{s}_{t+1}|\mathbf{s}_t, \mathcal{R}_t, \mathbf{a}_t^{\text{rule}})}_{P(A) = \int P(A,B) dB}$$

$$= \underbrace{P(\mathcal{R}_{t+1}|\mathbf{s}_{t+1}, \mathbf{s}_t, \mathcal{R}_t, \mathbf{a}_t^{\text{rule}}) \cdot P(\mathbf{s}_{t+1}|\mathbf{s}_t, \mathcal{R}_t, \mathbf{a}_t^{\text{rule}})}_{(c)\ P(\mathcal{R}_{t+1}|\mathbf{s}_{t+1}, \mathbf{s}_t, \mathcal{R}_t, \mathbf{a}_t^{\text{rule}}) = P(\mathcal{R}_{t+1}|\mathbf{s}_{t+1})}$$

$$= \underbrace{P(\mathbf{s}_{t+1}, \mathcal{R}_{t+1}|\mathbf{s}_t, \mathcal{R}_t, \mathbf{a}_t^{\text{rule}})}_{P(A|B,C) \cdot P(B|C) = P(A,B|C)}$$

$$= \underbrace{P(\tilde{\mathbf{s}}_{t+1}|\tilde{\mathbf{s}}_t, \mathbf{a}_t^{\text{rule}})}_{\tilde{s}_t := (\mathbf{s}_t, \mathcal{R}_t)}, \tag{9}$$

where $(a)$ follows from the fact that the transition to $\mathbf{s}_{t+1}$ is fully determined by current state $\mathbf{s}_t$ and current action to the environment $\mathbf{a}_t^{\text{env}}$, i.e., independent on rule set $\mathcal{R}_t$ and selected rule $\mathbf{a}_t^{\text{rule}}$; $(b)$ holds since $\mathbf{a}_t^{\text{env}}$ is determined only by the selected rule $\mathbf{a}_t^{\text{rule}}$ and the state $\mathbf{s}_t$; $(c)$ is due to our designed rule generation procedure where $\mathcal{R}_{t+1}$ is generated by the LLM from the the latest state $\mathbf{s}_{t+1}$. This completes the proof.

## E.2  PROOF OF PROPOSITION 5.1

We decompose the expected suboptimality into two cases:

$$\mathbb{E}_{\boldsymbol{\rho}_i \sim \pi_{\mathrm{LLM}}(\cdot|\mathbf{s})} \left[ Q^*(\mathbf{s}, \mathbf{a}^{\mathrm{env},*}(\mathbf{s})) - Q^*(\mathbf{s}, \mathbf{a}_i^{\mathrm{env}}) \right]$$
$$= \mathbb{P}\left[ \boldsymbol{\rho}_i \in \mathcal{R}^\delta(\mathbf{s}) \right] \cdot \mathbb{E}\left[ Q^*(\mathbf{s}, \mathbf{a}^{\mathrm{env},*}(\mathbf{s})) - Q^*(\mathbf{s}, \mathbf{a}_i^{\mathrm{env}}) \mid \boldsymbol{\rho}_i \in \mathcal{R}^\delta(\mathbf{s}) \right]$$
$$+ \mathbb{P}\left[ \boldsymbol{\rho}_i \notin \mathcal{R}^\delta(\mathbf{s}) \right] \cdot \mathbb{E}\left[ Q^*(\mathbf{s}, \mathbf{a}^{\mathrm{env},*}(\mathbf{s})) - Q^*(\mathbf{s}, \mathbf{a}_i^{\mathrm{env}}) \mid \boldsymbol{\rho}_i \notin \mathcal{R}^\delta(\mathbf{s}) \right]$$
$$\leq \mathbb{P}\left[ \exists i \in [N], \boldsymbol{\rho}_i \in \mathcal{R}^\delta(\mathbf{s}) \right] \cdot \mathbb{E}\left[ Q^*(\mathbf{s}, \mathbf{a}^{\mathrm{env},*}(\mathbf{s})) - Q^*(\mathbf{s}, \mathbf{a}_i^{\mathrm{env}}) \mid \boldsymbol{\rho}_i \in \mathcal{R}^\delta(\mathbf{s}) \right]$$
$$+ \mathbb{P}\left[ \forall i, \boldsymbol{\rho}_i \notin \mathcal{R}^\delta(\mathbf{s}) \right] \cdot \mathbb{E}\left[ Q^*(\mathbf{s}, \mathbf{a}^{\mathrm{env},*}(\mathbf{s})) - Q^*(\mathbf{s}, \mathbf{a}_i^{\mathrm{env}}) \mid \boldsymbol{\rho}_i \notin \mathcal{R}^\delta(\mathbf{s}) \right]. \tag{10}$$

When $\boldsymbol{\rho}_i \in \mathcal{R}^\delta(\mathbf{s})$, we have:
$$Q^*(\mathbf{s}, \mathbf{a}^{\mathrm{env},*}(\mathbf{s})) - Q^*(\mathbf{s}, \mathbf{a}_i^{\mathrm{env}}) \leq \delta.$$

When $\boldsymbol{\rho}_i \notin \mathcal{R}^\delta(\mathbf{s})$, we use:
$$Q^*(\mathbf{s}, \mathbf{a}^{\mathrm{env},*}(\mathbf{s})) - Q^*(\mathbf{s}, \mathbf{a}_i^{\mathrm{env}}) \leq \epsilon_{\mathrm{worst}}.$$

The probability that none of the sampled rules are in $\mathcal{R}^\delta(\mathbf{s})$ is:
$$\mathbb{P}\left[ \forall i, \boldsymbol{\rho}_i \notin \mathcal{R}^\delta(\mathbf{s}) \right] \leq (1 - \eta_s)^N.$$

Therefore, the overall expected suboptimality is bounded by:
$$\mathbb{E}\left[ Q^*(\mathbf{s}, \mathbf{a}^{\mathrm{env},*}(\mathbf{s})) - Q^*(\mathbf{s}, \mathbf{a}_i^{\mathrm{env}}) \right] \leq \delta + \epsilon_{\mathrm{worst}} \cdot (1 - \eta_s)^N. \tag{11}$$

This completes the proof.

### E.3 PROOF OF THEOREM 5.3

*Notation.* We will denote as $\mathcal{M}$ the original MDP as $\tilde{\mathcal{M}}$ the MDP for the rule-selection agent with transition function as in Theorem 4.1 and reward function $\tilde{R}(\tilde{\mathbf{s}}_t, \mathbf{a}_t^{\mathrm{rule}}) = R(\mathbf{s}_t, \mathbf{a}_t^{\mathrm{env}}) + \lambda R^{\mathrm{rule}}(\mathbf{a}_t^{\mathrm{rule}})$, where $\lambda \geq 0$ is a coefficient weight to balance the rule reward (typically very small) and $\mathrm{image}(R^{\mathrm{rule}}) = [0, 1]$. Throughout, we consistently use the tilde-notation $\sim$ for objects in $\tilde{\mathcal{M}}$ and non-tilde notation for objects in $\mathcal{M}$. Denote the optimal policy of $\mathcal{M}$ as $\pi^*$. By standard MDP arguments, we may assume that $\pi^*$ is deterministic. Recall that the states for $\tilde{\mathcal{M}}$ consist of states-rules pairs $\tilde{\mathbf{s}}_t := (\mathbf{s}_t, \mathcal{R}_t)$ and the LLM agent's actions are the selected rules $\mathbf{a}_t^{\mathrm{rule}}$. The environment action is determined as $\mathbf{a}_t^{\mathrm{env}} = \mathrm{LLM}(\mathbf{a}_t^{\mathrm{rule}}, \mathbf{p}_t)$ where $\mathbf{p}_t$ is the task-state prompt. The RBRL policy is trained to maximize the reward $\tilde{R}(\tilde{\mathbf{s}}_t, \mathbf{a}_t^{\mathrm{rule}})$.

Basically, we need to characterize
$$\mathrm{Gap}(T, s_0) := V_{\mathcal{M}}^{\pi^*, T}(\mathbf{s}_0) - V_{\mathcal{M}}^{\pi_{\mathrm{RBRL}}, T}(\mathbf{s}_0)$$
$$= \underbrace{V_{\mathcal{M}}^{\pi^*, T}(\mathbf{s}_0) - V_{\tilde{\mathcal{M}}}^{\pi_{\mathrm{RBRL}}, T}(\mathbf{s}_0)}_{Term_1} + \underbrace{V_{\tilde{\mathcal{M}}}^{\pi_{\mathrm{RBRL}}, T}(\mathbf{s}_0) - V_{\mathcal{M}}^{\pi_{\mathrm{RBRL}}, T}(\mathbf{s}_0)}_{Term_2},$$
where $\pi^*$ is the optimal polciy for original MDP $\mathcal{M}$ and $\pi_{\mathrm{RBRL}}$ is the optimal policy for the extended MDP $\tilde{\mathcal{M}}$.

First, $Term_1 \leq 0$. The reason is that $\mathcal{M}$ and $\tilde{\mathcal{M}}$ share the same transition dynamics and $\tilde{\mathcal{M}}$ has a higher reward function than $\mathcal{M}$, we always have the following inequality
$$V_{\mathcal{M}}^{\pi^*, T}(\mathbf{s}_0) \leq V_{\tilde{\mathcal{M}}}^{\pi^*, T}(\mathbf{s}_0) \leq V_{\tilde{\mathcal{M}}}^{\pi_{\mathrm{RBRL}}, T}(\mathbf{s}_0), \tag{12}$$
where the second inequality holds due to the fact that $\pi_{\mathrm{RBRL}}$ is optimal for $\tilde{\mathcal{M}}$.

Next, we bound $Term_2$. It is also very straightforward as the single policy $\pi_{\mathrm{RBRL}}$ on the two MDPs $\mathcal{M}$ and $\tilde{\mathcal{M}}$ will results in the same trajectories $(\mathbf{s}_0, \mathbf{a}_0; \mathbf{s}_1; \mathbf{a}_1; \dots)$. Hence it can be bounded as
$$Term_2 = \sum_{t=0}^{T-1} \gamma^t [\tilde{r}(\mathbf{s}_t, \mathbf{a}_t) - r(\mathbf{s}_t, \mathbf{a}_t)]$$
$$= \lambda \cdot \sum_{t=0}^{T-1} \gamma^t R^{\mathrm{rule}}(\mathbf{a}_t^{\mathrm{rule}}) \leq \lambda \sum_{t=0}^{T-1} \gamma^t$$
$$= \lambda \cdot \frac{1 - \gamma^{T+1}}{1 - \gamma}, \tag{13}$$

where the inequality comes from the fact that rule reward $R^{\text{rule}}(\mathbf{a}_t^{\text{rule}}) \leq 1, \forall t$. Therefore, we have

$$\text{Gap}(T, \mathbf{s}_0) \leq \lambda \cdot \frac{1 - \gamma^T}{1 - \gamma}.$$

This completes the proof.

## F  EXPERIMENT SETUP

All baselines were trained on 3 seeds for 2000 environment steps each.

### F.1  ENVIRONMENTS DETAILS

#### F.1.1  WEARABLE DEVICE ASSIGNMENT DOMAIN

The simulator for the Uganda domain is adapted from (Boehmer et al., 2024) with minor modifications to simplify the problem. This section provides an overview of the environment, with additional details available in the original paper. In this environment, they want to allocate vital sign monitoring devices to mothers arriving in a maternal unit in order to better monitor mothers' health condition. Each mother's state is modeled by her vital signs (heart rate, respiratory rate, and blood oxygen saturation) along with each vital sign's variability. The mother's vital sign transition is governed by a multivariate Gaussian distribution defined over her vital signs at the current timestep and next timestep, learned from de-identified vital sign data collected from patients at Mbarara Regional Referral Hospital. MIMIC-III Johnson et al. (2016) is another de-identified clinical vital sign dataset that includes the same set of vital signs as the Uganda domain. The key difference is they have different data sources, as MIMIC-III's data comes from Beth Israel Deaconess Medical Center in Boston.

Wearing a monitoring device does not directly alter a mother's vital sign trajectory but has an indirect positive effect by triggering alerts when vital signs deviate from the normal range. These alerts often lead to medical interventions that improve the mother's condition. If no monitoring device is assigned (passive action), the mother's next state is sampled from the multivariate Gaussian distribution conditioned on the current state. If a monitoring device is assigned and the vital signs remain within the normal range, the vital signs evolve as under the passive action. However, if any vital sign deviates from the normal range, there is a 30% chance the vital signs evolve as under the passive action, based on empirical evidence suggesting that clinicians fail to respond in such cases 30% of the time (Boatin et al., 2021). Otherwise, vital signs are probabilistically adjusted towards the normal range before sampling the next state, modeling the positive impact of medical intervention.

The algorithm's goal is to optimize monitoring device allocation to maximize the aggregate reward across all mothers. We simplify the problem by requiring exactly one new mother to join the maternal unit at each timestep, starting with a single mother in the unit. The system has a budget of five monitoring devices. A device must be allocated to the new mother, and if all devices are already in use, one must be removed from a current user. Once removed, a device cannot be reassigned to the same mother. Each mother remains in the maternal unit for 10 timesteps, after which her vital sign trajectory no longer contributes to the reward. Once a device is taken from a mother, we directly sample her entire vital sign trajectory under passive action for the remaining timesteps she stays in the maternal unit and compute all her future rewards. We can directly compute future rewards because the mother will not receive the device again, so she will only undergo passive action in the remaining time. This observation enables us to maintain a smaller observation space, as we only need to keep track of the states of the mothers who own the device.

In this domain, the constraints can be written as $\|\mathbf{a}_t \in \mathcal{R}^{d_2}\|_1 \leq B, \forall t$, which $d_2$ represents the number of patients in the system at each time slot, and the 1-norm of the action vector must remain within the budget $B$.

#### F.1.2  HEAT ALERTS DOMAIN

The heat alert issuance problem can be modeled as an MDP in the context of RL Considine et al. (2025). The state at any given time, denoted as $\mathbf{s}_t$, encompasses both exogenous and endogenous factors. Exogenous factors include current weather conditions, such as the heat index, temperature,

and humidity, which directly influence the risk of heat-related health issues. Endogenous factors include the history of issued alerts, such as the number and timing of past alerts, their effectiveness, and the remaining budget for the season. Additionally, the day of the week is considered, as public responsiveness to alerts may vary between weekdays and weekends. The action space is binary, with $\mathbf{a}_t \in \mathbb{Z}_2$. The decision to issue a heatwave alert $\mathbf{a}_t = 1$ or not $\mathbf{a}_t = 0$ is constrained by the remaining alert budget. If the budget is exhausted, no further alerts can be issued. The reward function is designed to reflect the reduction in heat-related hospitalizations, which depends on the effectiveness of the alert under current weather conditions. A Bayesian hierarchical framework could be employed to model the health impact of alerts, capturing the uncertainty in their effectiveness. Importantly, consecutive alerts tend to lose effectiveness, introducing a diminishing returns effect that must be accounted for in the decision-making process.

The transition dynamics, $P(\mathbf{s}_{t+1}|\mathbf{s}_t, \mathbf{a}_t)$, describe how the system evolves over time. The next state is influenced by weather trajectories, the action taken, and public responsiveness to alerts. For instance, issuing an alert reduces the remaining budget and updates the history of issued alerts, while the weather conditions may change independently. Public responsiveness may also vary based on the frequency and timing of past alerts. A key constraint in this problem is the limited alert budget, which necessitates strategic allocation of alerts throughout the season. The goal is to learn a policy $\pi(\mathbf{s}_t|\mathbf{a}_t)$ that maximizes cumulative rewards by effectively issuing alerts during severe heat conditions to minimize hospitalizations, while conserving the budget for future use. This involves balancing immediate health benefits against the potential need for alerts later in the season, addressing the trade-offs between short-term and long-term outcomes.

For this domain, the budget constraints can be expressed as $\sum_{t=1}^{h} \mathbf{a}_t \in \mathbb{R} \leq B$, where the total sum of all actions over time horizon $h$ must not exceed a budget threshold $B$.

### F.1.3 BIN PACKING DOMAIN

We adopt the online stochastic bin packing environment from Balaji et al. (2019), which models the sequential allocation of items with random sizes into fixed-capacity bins. At each time step $t$, an item $x_t \in (0, 1)$ is sampled from a known distribution and must be immediately placed into one of the currently open bins or into a new bin. Each bin has a fixed capacity $B$, and assigning an item to a bin must satisfy the constraint that the total usage does not exceed $B$. The environment is formulated as MDP where state $\mathbf{s}_t$ is a tuple containing the size of the arriving item $x_t$ and the current fill levels of all open bins, action $\mathbf{a}_t$ selects a valid bin from the set of open bins or opening a new bin; invalid actions (bins that would overflow) are masked. The transition probablity $P(\mathbf{s}_{t+1}|\mathbf{s}_t, \mathbf{a}_t)$ determines update of the selected bin's fill level, followed by stochastic sampling of the next item $x_{t+1}$. The reward function $r_t$ is defined as the negative incremental waste after placing item $x_t$, i.e., the change in unused space across all open bins. The objective is to learn a policy $\pi(\mathbf{a}_t|\mathbf{s}_t)$ that minimizes the total accumulated waste over time. This domain reflects a broad class of real-world resource allocation settings-such as server packing, memory management, or warehouse logistics, where agents must make immediate, irrevocable decisions under hard capacity constraints.

For this domain, the budget constraints can be expressed as $x_t + C_{\mathbf{a}_t} \leq B$, where at every time no bin can exceed the capacity $B$, with $C_{\mathbf{a}_t}$ being the current usage of bin $\mathbf{a}_t$.

### F.2 GYM ENVIRONMENTS AN LANGUAGE WRAPPERS

We implemented the `WearableDevicesAssignment` environments as gymnasium environments Towers et al. (2024), while the `HeatAlerts` domain was already available in this format. We additionally created a `LanguageWrapper` Python class described in Table 2, which can be applied to any `gymnasium` environment. Our code implementations can be applied to any environment wrapped in this class.

### F.3 RL IMPLEMENTATIONS, HYPERPARAMETERS AND SETTINGS

We implemented three main RL algorithms for the experiment sections: Attention-based SAC for `RBRL`, numeric PPO, and Finetuning-based PPO. We based our implementation on the single-file, high-quality implementations from the `cleanrl` project (Huang et al., 2022). For Attention-based SAC, we required significant changes to keep track of the rule-augmented state space, as described in

| Method/Property | Type | Description |
|---|---|---|
| `task_text` | Property (Abstract) | Returns a description of the task that the environment is solving. |
| `action_space_text` | Property (Abstract) | Returns a description of the action space of the environment. |
| `state_descriptor(obs, info)` | Abstract Method | Converts the observation into a text description. |
| `step(action)` | Method | Wraps the step method of the environment adding the text representation to the state space. |
| `reset(seed, options)` | Method | Wraps the reset method of the environment adding the text representation to the state space. |
| `action_parser(s)` | Method | Parses an action string and converts it into an appropriate format for the environment's action space. |
| `(rule_examples` | Property (Optional) | Returns a list of string representation of rules. |

Table 2: Methods and properties of the `LanguageWrapper` class

Section 4.4. Other major changes to the baseline SAC implementation (originally designed for Atari) were more frequent target network updates and updating the actor and critic four times per iteration. This was done to improve sample efficiency and cope with the slow generation by the LLM. Numeric PPO was used practically without modification.

For the Finetuning-based PPO, we used low-rank adaptation (LoRA) Hu et al. (2021) with the Transformers package and models hosted on Llama Hugging Face Wolf et al. (2020). We set the rank to $r = 1$ and the adaptation weight to 2, resulting in only 0.8% trainable parameters (still an order of magnitude larger than the Attention-based policy). Tables 3, 4, and 5 show the hyperparameters and settings used in these implementations.

## F.4 COMPUTING ENVIRONMENT

SAC attention can run on a regular laptop since most of the computation happens in the cloud through API LLM calls, while the RL module is small and can run on personal CPUs. Nonetheless, the process is bottlenecked by the speed of generation from the APIs. A full run of 2 million environment steps, with parallelized API calls across four environments, took approximately four hours to complete. One training cycle did not exceed $10 in API costs. However, all the experiments and development incurred approximately $1,500 in API costs. As described in the main text, the LLM fine-tuning experiments used an Nvidia A100 40GB GPU for each seed, training on three seeds for 18 hours each. Computations were performed on a Slurm-based high-performance computing cluster.

Table 3: SAC Hyperparameters and Settings for `RBRL`.

| Parameter | Default Value | Description |
|---|---|---|
| `num_envs` | 4 | Number of parallel environments |
| `total_timesteps` | 500 | Total number of environment steps |
| `gamma` | 0.95 | Discount factor $\gamma$ |
| `tau` | 1.0 | Target smoothing coefficient |
| `batch_size` | 16 | Batch size of sample from the replay memory |
| `buffer_size` | 4096 | The replay memory buffer size |
| `max_episode_steps` | 32 | Episode truncation |
| `learning_starts` | 256 | Timestep to start learning |
| `policy_lr` | $1 \times 10^{-4}$ | Learning rate of policy network optimizer |
| `q_lr` | $1 \times 10^{-4}$ | Learning rate of Q-network optimizer |
| `actor_updates` | 4 | Number of actor updates per update cycle |
| `critic_updates` | 4 | Number of critic updates per update cycle |
| `target_network_frequency` | 64 | The frequency for the target network update |
| `alpha` | 0.01 | Initial entropy regularization coefficient |
| `autotune` | True | Automatic tuning of the entropy coefficient |
| `target_entropy_scale` | 0.89 | Coefficient for scaling the autotune entropy target |
| `dropout` | 0.05 | The dropout rate |
| `num_rules` | 10 | Number of rules for `RBRL` |
| `llm` | "gpt-4o-mini" | LLM for generation |
| `embedder_lm` | "m2-bert-80M-8k-retrieval" | The LLM to use for embeddings |
| `embed_dim` | 768 | Dimension of rule embeddings |
| `hidden_dim` | 16 | Hidden dimension of networks |
| `rule_reward_coef` | 0.1 | The reward coefficient for the rules |
| `num_self_attention_layers` | 1 | For the actor and critic |
| `num_cross_attention_layers` | 1 | For the actor and critic |

Table 4: Numeric PPO Hyperparameters and Settings.

| Parameter | Default Value | Description |
|---|---|---|
| total_timesteps | 50000 | Total timesteps of the experiments |
| learning_rate | $2.5 \times 10^{-4}$ | Learning rate of the optimizer |
| num_envs | 4 | Number of parallel environments |
| num_steps | 512 | Steps per policy rollout |
| anneal_lr | False | no learning rate annealing |
| gamma | 0.95 | Discount factor $\gamma$ |
| gae_lambda | 0.95 | Lambda for Generalized Advantage Estimation |
| num_minibatches | 4 | Number of mini-batches |
| update_epochs | 64 | Number of update epochs |
| norm_adv | True | Whiten advantages |
| clip_coef | 0.2 | Surrogate clipping coefficient |
| clip_vloss | True | Clipped loss for value function |
| ent_coef | 0.01 | Coefficient of entropy term |
| vf_coef | 0.5 | Coefficient of value function |
| max_grad_norm | 0.5 | Maximum gradient clipping norm |
| target_kl | None | Target KL divergence threshold |
| hidden_dim | 16 | Hidden dimension of networks |
| num_hidden_layers | 2 | For policy and critic networks |
| max_episode_steps | 32 | Episode truncation |

Table 5: LLM PPO Finetuning Hyperparameters and Settings.

| Parameter | Default Value | Description |
|---|---|---|
| total_timesteps | 500 | Total number of timesteps |
| learning_rate | $2.5 \times 10^{-4}$ | Learning rate of optimizer |
| num_envs | 4 | Number of parallel game environments |
| num_steps | 32 | Steps per policy rollout |
| anneal_lr | True | Enable learning rate annealing |
| gamma | 0.95 | Discount factor $\gamma$ |
| gae_lambda | 0.95 | Lambda for Generalized Advantage Estimation |
| update_epochs | 4 | Number of update epochs per cycle |
| norm_adv | True | Advantages whitening |
| clip_coef | 0.2 | Surrogate clipping coefficient |
| clip_vloss | True | Clipped loss for value function |
| ent_coef | 0.01 | Coefficient of entropy term |
| vf_coef | 0.5 | Coefficient of value function |
| kl_coef | 0.05 | KL divergence with reference model |
| max_grad_norm | 0.5 | Maximum gradient clipping norm |
| target_kl | None | Target KL divergence threshold |
| dropout | 0.0 | Dropout rate |
| llm | "meta-llama/Llama-3.1-8B-Instruct" | Model to fine-tune |
| train_dtype | "float16" | Training data type |
| gradient_accumulation_steps | 16 | Number of gradient accumulation steps |
| minibatch_size | 1 | Mini-batch size for fine-tuning |
| max_chunk_size | 256 | Maximum length sequence for the back propagation |
| max_episode_steps | 32 | Maximum number of steps per episode |

# G  ADDITIONAL EXPERIMENTAL RESULTS

## G.1  RBRL + ADVANCED REASONING LLM

We conducted a new experiment using the advanced reasoning baseline "o3-mini" from ChatGPT, implemented within our MimicIII environment. This baseline has been trained with RL to solve reasoning and mathematical problems more effectively. The results are shown in Table 6 below.

Here, we observe that using RBRL with the advanced reasoning baseline helped to almost match the performance of the Oracle policy in very few environment steps, illustrating the compatibility of RBRL with reasoning-based LLMs. However, we remark that relying on o3-mini resulted in almost 10x higher cost and 5x slower inference than 4o-mini used for most experiments. As noted in the manuscript, our total cost for the experiments was approximately 2000, which would increase greatly with o3-mini. Hence, we focused on improving cost-effective LLMs.

Table 6: RBRL + Advanced Reasoning on MimicIII (Av. Reward)

| Algorithm | @500 steps | @2.5k steps | @100k steps (working oracle) |
|---|---|---|---|
| PPO | $-0.66 \pm 0.12$ | $-0.55 \pm 0.07$ | $-0.14 \pm 0.06$ |
| SAC | $-0.85 \pm 0.12$ | $-0.81 \pm 0.15$ | $-$ |
| RBRL (GPT–4o mini) | $-0.70 \pm 0.07$ | $-0.33 \pm 0.03$ | $-$ |
| RBRL+ AdvancedReasoning (GPT–o3 mini) | $-0.29 \pm 0.05$ | $-0.16 \pm 0.05$ | $-$ |

## G.2 COMPARISON OF OTHER LLM FINETUNING METHODS FOR MDPS

In the main paper, we compared RBRL with token-wise LLM finetuning using LORA in the MimicIII domain. We further implemented the methodology described by Zhai et al. (2024) which provides a variation of the finetuning PPO methodology in which the log probability of an action is first aggregated across all tokens and no reference model reward (KL) divergence is used. The authors also introduce a weight coefficient $\lambda_{cot}$ to reduce the weight of the COT tokens in the log-probability computation. The results are shown in Table 7. The best method was still RBRL, while the second best was the finetuning baseline at the action level with $\lambda_{cot} = 1.0$.

Table 7: New baseline on MIMICIII with Llama 3.1 8B* (Av. Rewardx

| Baseline | Av. Return @ 2.5k |
|---|---|
| Finetuning – token level | $-0.85 \pm 0.21$ |
| Finetuning – action level ($\lambda_{cot} = 1.0$) | $-0.46 \pm 0.09$ |
| Finetuning – action level ($\lambda_{cot} = 0.3$) | $-0.90 \pm 0.12$ |
| **RBRL***  | $\mathbf{-0.36 \pm 0.05}$ |

## G.3 CONSISTENCY OF EXPLANATIONS

A central claim of our work is that RBRL generates consistent explanations. Following the discussion in the literature (Lyu et al., 2024; Jacovi & Goldberg, 2020), we define a consistent explanation as one that is causally linked to the decision process. In RBRL, this consistency is achieved by design through a strict, decomposable pipeline: (`observation` → `rule` → `action`). Our key argument, which we evaluate empirically below, is that the state and the selected rule alone are sufficient to determine the action taken, regardless of the intermediate reasoning trace used to generate the rule. The rule itself serves as the true, functional explanation for the agent's decision.

**Action selection comes from the rule, not the reasoning trace** We show that including or not including the reasoning trace does not alter the action decision, which is based on the rule. For this purpose, we use a pretrained RBRL policy to collect tuples (`state, reasoning, rule, action`). Next we evaluate how much the action selection changes when including the reasoning trace in addition to the rule in the action prompt.

**Metrics.** We compute the agreement score as the fraction of times that the action selected without including the LLM reasoning in the action prompt is the same as that when including it. A higher value means the rule determines the action, not the LLM reasoning output. All experiments are with `gpt-4o-mini` as in the main paper. The scores are averaged over 100 environment steps with standard errors.

Table 8: Consistency of action selection

| Environment | Agreement score $\pm$ SE |
|---|---|
| HeatAlerts | $0.99 \pm 0.010$ |
| Uganda | $0.92 \pm 0.027$ |
| Mimic III | $0.92 \pm 0.027$ |
| BinPacking | $0.97 \pm 0.017$ |

Our results confirm that the action consistency comes mainly from the state and rule, not the LLM reasoning, further showing that LLM self-explanation is not the core of RBRL. In terms of Jacovi & Goldberg (2020), RBRL is more closely related to the class of methods where, *"the explanation [rule] is itself a model capable of making decisions (e.g., decision trees or rule lists)."*

### G.4 EVALUATING THE TRANSLATION FIDELITY FROM RULES TO EXPLANATIONS

Having established in our previous experiment that the rule is the true causal component of the decision, a critical next question arises: Does the final, user-facing natural language explanation translate all decision-relevant information from that rule? An ideal explanation should suffer no information loss in this translation, meaning it should be self-contained and sufficient for an observer to deduce the action that was taken.

To verify this, we conduct an action-prediction experiment. We provide an auxiliary LLM with the agent's full explanation but mask the final action (e.g., "...Thus, I concluded XXX."). The LLM's task is to predict the masked action. High prediction accuracy serves as a strong proxy for high-fidelity translation, indicating that the explanation successfully preserves the action-predictive information from the original rule.

An example of the experimental setup is shown in Table 9. Figure 7 presents the results of this evaluation across three domains: `HeatAlerts`, `Uganda`, and `MimicIII`. The plot compares the classification accuracy and F1 score of a binary classifier trained to predict the agent's actions solely from the masked explanations. We benchmark `RBRL` against two baselines: `CoT`, which produces naive rationale via chain-of-thought prompting, and `TBRL`, which incorporates optimization over toughts. Across all domains, `RBRL` consistently outperforms both baselines with higher accuracy and F1 scores. In particular, the large margin between `RBRL` and `CoT` highlights the importance of grounding explanations in decision-consistent rules rather than free-form text. The error bars represent 95% confidence intervals obtained via bootstrapping, demonstrating statistical robustness. These results confirm that `RBRL`'s explanations are more than just plausible narratives; they are reliable translations of the agent's underlying decision mechanism, preserving enough information to be considered truly consistent and trustworthy.

Table 9: Examples of Masked Explanations (HeatAlerts)

| Agent | Explanation |
|---|---|
| CoT Reasoning | I observed a heat index of 100 F, a warning streak of 6, and a remaining budget for 2 warnings. Forecast shows high temperatures next week. I concluded XXX. |
| RBRL | I observed a heat index of 100 F with 6 warnings in 14 days. Reasoning that excessive warnings lead to fatigue, I applied a rule to not issue warnings if warnings in the last 7 days $\geq 5$ and budget remained. Thus, I concluded XXX. |

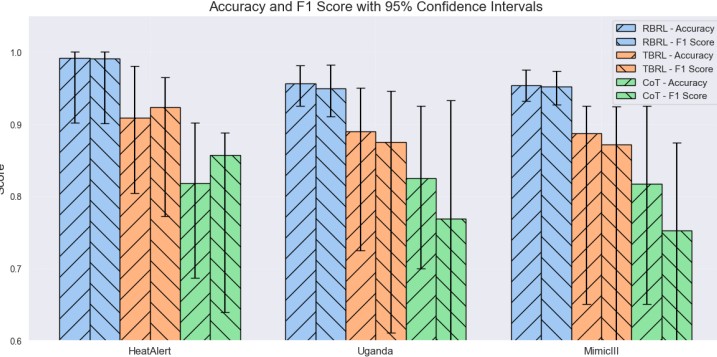

Figure 7: Consistency of explanation as measured by the action prediction task.

## G.5 Study on Rule Diversity

As noted in Section E, we only require that at least one candidate rule yields the optimal actions with high probability; non-optimal actions are irrelevant. We conducted an experiment to assess this assumption in practice.

**Setup.** We use an auxiliary oracle numeric SAC policy trained for 100k steps per environment. The metric is the percentage of times the proxy optimal action is recommended by at least one rule. The experiment is repeated for 100 transitions.

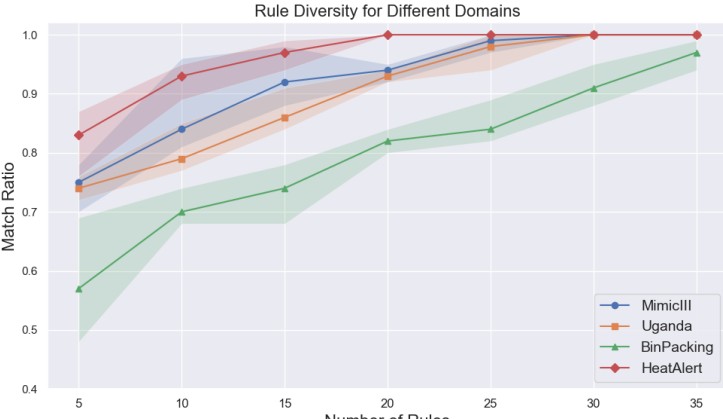

Figure 8: Rule diversity verification.

Figure 8 illustrates the impact of increasing the number of rules on the match ratio between rule-based decisions and the reference RL agent actions across four environments: `HeatAlert`, `MimicIII`, `Uganda`, and `BinPacking`. The match ratio serves as a proxy for rule-to-action consistency, reflecting how well the selected rules align with the RL agent's behavior. Across all environments, increasing the number of candidate rules improves the match ratio, indicating that greater rule diversity enhances the likelihood of covering the correct action. The shaded regions represent variance across multiple runs, highlighting robustness trends in each domain. *In summary, as the number of rules generated increased, the probability of generating an optimal rule candidate increased, thereby increasing the performance guarantees of* `RBRL` *as discussed in section 5.*

## G.6 Additional results for General RL domain

Table 10: Average Return on general RL domain: BabyAI

| Baseline | BabyAIGotoObj | BabyAIGotoLocal |
|---|---|---|
| CoT (GPT-4o mini) | $0.12 \pm 0.12$ | $0.12 \pm 0.17$ |
| RBRL* @ 5k steps (GPT-4o mini) | $\mathbf{0.46 \pm 0.17}$ | $\mathbf{0.32 \pm 0.11}$ |
| SAC @ 5k steps | $0.27 \pm 0.22$ | $0.16 \pm 0.18$ |

Besides the three aforementioned resource-constrained allocation domains, we further evaluate our `RBRL` in the more general standard RL settings. We applied our methodology directly to the widely used BabyAI domain Chevalier-Boisvert et al. (2018). The environment was originally proposed to investigate sample efficiency and generalization in RL. We run on two environments "BabyAIG-otoObj", "BabyAIGotoLocal" environments using the standard wrapper for full observability are included in the official wrapper in the minigrid Python library. Improved performance for these two general RL domains can be found in Table 10. Numeric RL baselines can perfectly solve these tasks albeit requiring potentially millions of observations Chevalier-Boisvert et al. (2018). Here we focus on a small sample regime. We do not position our contribution as a general method for RL and nor claim any superiority over state of the art general-purpose RL.

# H ADDITIONAL SURVEY RESULTS

Figure 9 illustrates the results of a human survey conducted to evaluate the quality of explanations generated by our method compared to alternatives. A total of 21 valid responses were collected for the `HeatAlert` environment (Figure 9a), and 20 valid responses were gathered for the `Uganda` environment (Figure 9b). As shown in the figures, our method was favored by the majority of participants across all cases. In the `HeatAlert` environment, the preference for our approach is evident, although there is a small percentage of tied and "Not Preferred" responses. In contrast, the preference for our method is even more pronounced in the `Uganda` environment, with a significantly higher number of participants selecting "Ours Preferred." These results demonstrate the effectiveness of our approach in generating explanations that resonate better with human users, particularly in the `Uganda` domain.

Figure 10 illustrates the survey outcomes obtained by querying LLMs 20 times for each case in the `HeatAlerts` (Figure 10a) and `Uganda` (Figure 10b) environments. To ensure variability, the LLM's sampling temperature was controlled, enabling randomized responses for each trial. Similar to the human survey results, our method ("Ours Preferred") is overwhelmingly favored across all cases in both domains. Notably, the consistency of "Ours Preferred" responses highlights the effectiveness of our approach in generating explanations that align well with the LLM's evaluation criteria, further validating the robustness of our methodology.

Figure 11 illustrates the survey results evaluating hallucination occurrences across two environments (`Uganda` and `HeatAlert`) for three explanation types: Chain of Thought (`CoT`), Rule-Bottleneck Reinforcement Learning (`RBRL`), and `None` (indicating no explanation).

In Figure 11a, the results from the human survey indicate that `CoT`-based explanations had a significant proportion of hallucinations, particularly in the `Uganda` environment, where it accounted for 42.4% of responses. `RBRL` explanations showed markedly fewer hallucinations in both domains, highlighting its robustness. A notable percentage of responses for `None` indicate scenarios where explanations were either absent or irrelevant. In Figure 11b, results from the LLM survey further emphasize the trends observed in the human survey. Hallucination rates for `CoT` were even higher in the `Uganda` environment (81.7%), whereas `RBRL` explanations exhibited almost no hallucinations across both domains. In the `HeatAlert` environment, the absence of explanations (`None`) led to the highest percentage of hallucinations, underlining the importance of well-structured, rule-based explanations like `RBRL`. Need to mention that the plausibility (human preference) of explanations of `RBRL` shown in the human survey was not conditional on seeing the CoT reasoning traces. These results collectively demonstrate that the `RBRL` framework significantly mitigates hallucinations, providing more accurate and reliable explanations compared to other methods.

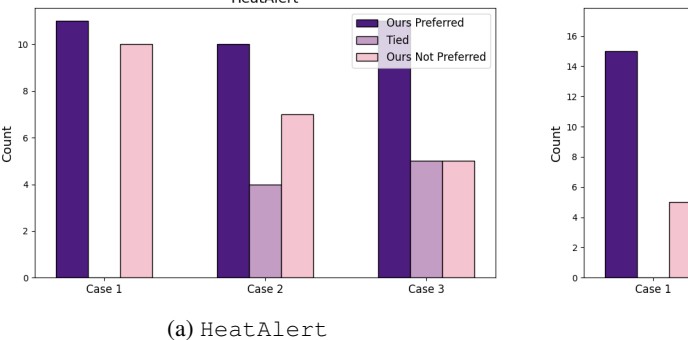
(a) `HeatAlert`

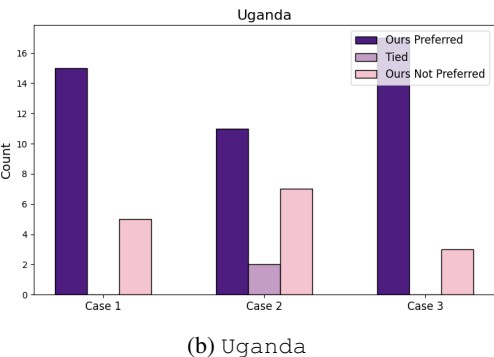
(b) `Uganda`

Figure 9: Results from **human** surveys conducted in the `HeatAlert` (a) and `Uganda` (b) environments. 21 participants provided feedback for the `HeatAlert` domain, while 20 valid responses were collected for the `Uganda` domain. The results indicate that our method ("Ours Preferred") was favored by a majority of participants, particularly in the `Uganda` domain, where the preference is more pronounced.

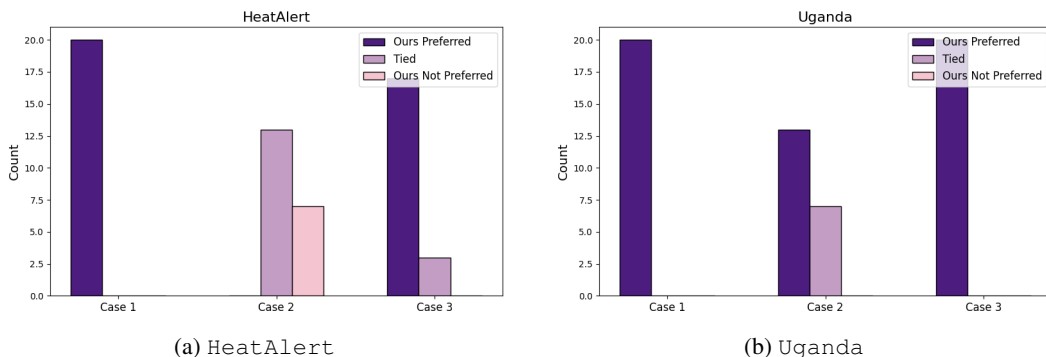

(a) `HeatAlert`                                    (b) `Uganda`

Figure 10: Survey results generated by querying **LLMs** 20 times for each case in the `HeatAlert` (a) and `Uganda` (b) environments. By varying sampling temperatures, randomized responses were collected. The results show that our method ("Ours Preferred") consistently outperforms alternatives across all cases, highlighting its robustness and alignment with the evaluation criteria of the LLMs.

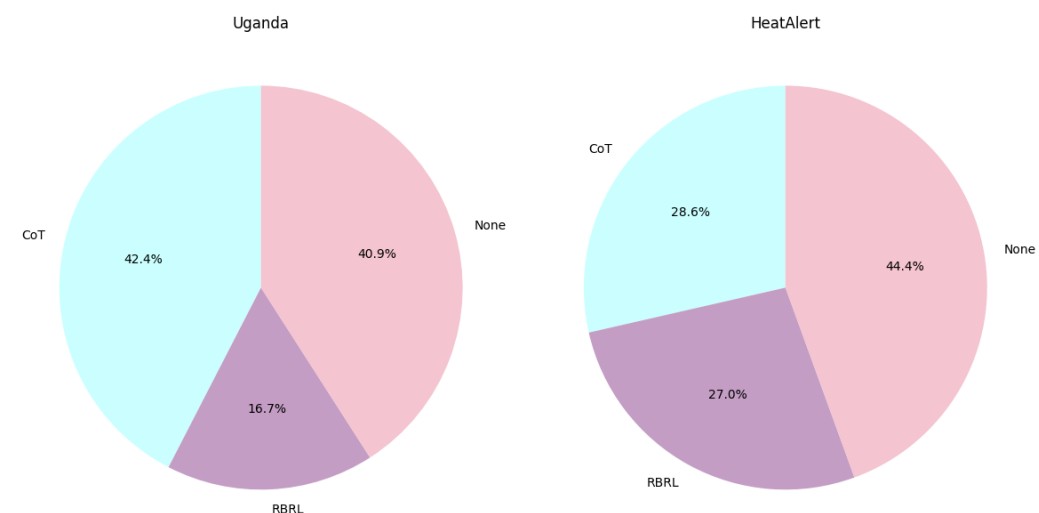

(a) Results from the **human** survey, showcasing the proportion of hallucination detected across three categories: `CoT`, `RBRL`, and `None`. In both domains, hallucinations were most frequently identified in `None`, with `RBRL` showing significantly fewer instances.

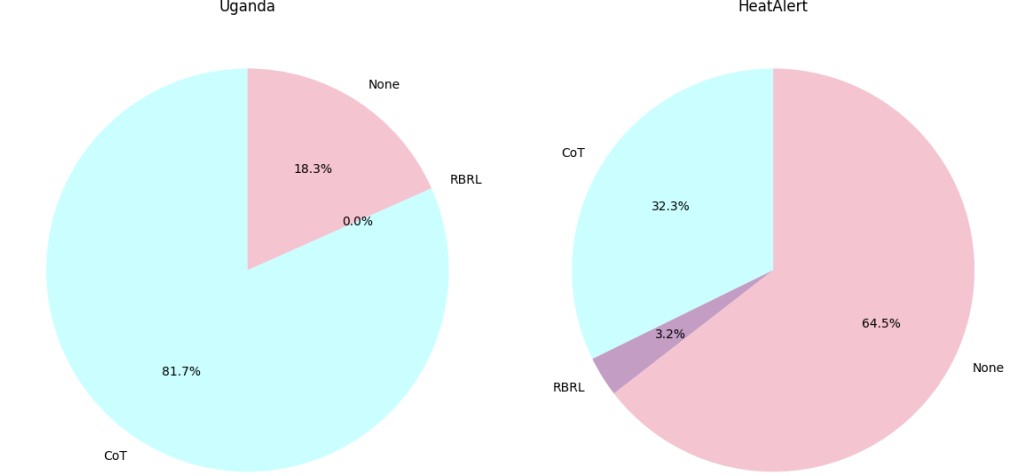

(b) Results from the **LLM**-based survey, where hallucination detection was assessed through multiple iterations of LLM evaluation. `CoT` exhibited higher hallucination rates in the `Uganda` domain, while `RBRL` demonstrated minimal hallucination occurrences in both domains.

Figure 11: Survey results for hallucination detection across the `HeatAlert` and `Uganda` environments.

# I PROMPT TEMPLATES AND RULE EXAMPLES

## I.1 PROMPT FORMAT

In this section, we illustrate the prompt format used in our `RBRL` for generating thoughts, rules, actions, rule scores, and explanations in Figure 12.

---

### Generate Thoughts

First, reason about what elements should be considered when choosing the optimal action. Your response should consist of a single short paragraph that reflects on the consequences, benefits, and drawbacks of each action in the current state.

---

### Generate Rules

Now, suggest {num_rules} rules that could be useful to make an optimal decision in the current state. For each rule, provide the explanation of why it is important to consider it at the given state. Each rule should be in machine-readable JSON Lines format. Each line should follow the following schema:

{'background' str, 'rule': str, 'state relevance': str, 'goal relevance': str}

- The 'background' should a brief introduction and motivation to the focus of the rule.
- The 'rule' should be a statement of the form '[do/select/prioritize] [if/when/condition]' where the condition must be relevant to the current state.
- The 'state relevance' should explain why the rule applies to the current problem state.
- The rule alone should be sufficient to deduce the optimal action that should be taken in the current problem state.
-   Start each line with the character '```- {.

---

### Generate Actions

Below is/are a prioritization rule/rules to make an optimal decision in the current state:

[selected rule/rules]

Now, choose the optimal action given the current problem state and this/these prioritization rule/rules. Your answer must consist exclusively of one of the following actions:

Possible actions:{description about action space}

You cannot refuse to respond. Do not provide additional information or context for your answer, only the action.

---

### Generate Rule Scores

To make a decision in the current state, the following rule/rules was/were selected:

{rules}

You will now be given a question you need to answer with a simple 'yes' or 'no'.

q1 = "Is/are the rule/rules **alone** sufficient to understand the optimal action/decision that the system should take in current the problem state?"

q2 = "Is the condition in the rule/rules actionable and complete in the current problem state (containing sufficient detail about the current problem state without unnecessary information)?"

q3 = "Did the selected rule/rules sufficiently help to understand the previous decision without contradictions?"

Answer the following questions with a simple 'yes' or 'no' without additional information or justification. Your response should be a single word.

---

### Generate Explanation

You chose action {outputs['action']} in the current problem state. Explain why you chose the optimal action based on the conversation and history. Your response should follow the template: "I observed... I used a rule stating... I concluded..."

---

Figure 12: Prompts template for generating thoughts, rules, actions, rule scores, and explanations.

## I.2   RULE EXAMPLE

In this section, we provide some rule examples for each domain in Figure 13.

---

### Rule Examples

#### Example 1: Wearable Device Assignment

- **Rule1**: "Prioritize reallocating devices from patients who have worn the device for the least amount of time and have stable vital signs."
- **Rule2**: "Select to reallocate devices from patients with less critical vital signs to those with high pulse rates or low SPO2."
- **Rule3**: "Reallocate devices from patients whose mean vital signs are within normal ranges to those whose mean vital signs are abnormal."
- **Rule4**: "Select to reallocate devices from patients with lower standard deviation in vital signs, as they are less likely to experience sudden changes."
- **Rule5**: "When a new patient arrives, reallocate devices from patients who have shown the least improvement over time."

#### Example 2: Heat Alert Issuance

- **Rule1**: "If the current heat index is above the average heat index of the past week, issue a warning."
- **Rule2**: "If there have been 5 or more warnings in the last 14 days, do not issue a warning unless the heat index exceeds 100 F."
- **Rule3**: "If the heat forecast for the next day exceeds 98 F, issue a warning today."
- **Rule4**: "If today is a weekday and the heat index is above 95 F, do not issue a warning unless the forecasted heat index for the weekend exceeds 100 F."
- **Rule5**: "If the remaining number of warnings is 5 or fewer, prioritize issuing a warning when the heat index exceeds 98 F."

#### Example 3: Bin Packing Allocation

- **Rule1**: "If the current item is less than half the bin capacity, evaluate existing bins first before creating a new one."
- **Rule2**: "When possible, open a new bin instead of using a level that has already been used heavily."
- **Rule3**: "Select a bin such that the remaining space is closest to the bin capacity after placing the item."
- **Rule4**: "Always check if any bins are completely empty before making any decisions."
- **Rule5**: "If any bin can accommodate the item exactly without waste, prioritize that choice."

Figure 13: Rule examples for the considered two domains.

## J  SURVEY EXAMPLE

In this section, we present a survey example from the `Wearable_Device_Assignment` domain. The survey for the `HeatAlert` domain follows the same format and can be easily adapted by substituting the task and corresponding actions. For brevity, we include only one example case from the `Wearable_Device_Assignment` domain.

**Task:** *You are tasked with optimizing the allocation of limited vital sign monitoring devices among patients. Devices improve vital signs and prevent abnormalities, but their limited availability requires reallocating them from stable patients to higher-risk incoming patients, who must always receive a device. The normal range of vital signs are provided in Figure 14. The goal is to minimize costs associated with abnormal vital signs, where costs are calculated exponentially based on deviations from predefined thresholds. Wearing a device improves abnormal vital signs with a 70% success rate.*

**Possible actions:** *Choose the id of the device that will be reallocated to the new incoming patient. Your answer should be a single integer i from 0 to 4 (the number of devices) such that:*

- *Always choose a free device if available*

- *If no free device is available, then choose device i whose current patient is at least risk or would benefit less from wearing the device.*

| Vital Sign | Normal Range |
|---|---|
| Heart Rate | 60–100 bpm |
| Respiratory Rate | 12–20 breaths/min |
| SPO2 (Oxygen Saturation) | ≥ 95% |

Figure 14: Normal range of vital signs.

*In the following, you will be presented with three cases. Each case includes two explanations. Please read the text for each case carefully and answer the questions provided.*

**Case 1:** Current state of the decision problem:
Number of devices: 5
Number of free devices: 1
IDs of free devices: 3

**Device 0:** Device is currently assigned to a patient with the following description:
*Timesteps wearing the device*: 1
*Pulse rate* - Last value: 95.22 - Mean: 105.12 - Standard deviation/volatility: 10.56
*Respiratory rate* - Last value: 20.14 - Mean: 20.54 - Standard deviation/volatility: 0.64
*SPO2* - Last value: 98.42 - Mean: 97.89 - Standard deviation/volatility: 0.88

**Device 1:** Device is currently assigned to a patient with the following description:
*Timesteps wearing the device*: 1
*Pulse rate* - Last value: 89.55 - Mean: 81.86 - Standard deviation/volatility: 8.55
*Respiratory rate* - Last value: 14.85 - Mean: 20.81 - Standard deviation/volatility: 3.40
*SPO2* - Last value: 95.31 - Mean: 96.22 - Standard deviation/volatility: 1.38

**Device 2:** Device is currently assigned to a patient with the following description:
*Timesteps wearing the device*: 1
*Pulse rate* - Last value: 106.05 - Mean: 105.09 - Standard deviation/volatility: 2.91
*Respiratory rate* - Last value: 19.34 - Mean: 20.80 - Standard deviation/volatility: 2.69
*SPO2* - Last value: 99.56 - Mean: 99.36 - Standard deviation/volatility: 0.27

**Device 3:** Device is currently free.

**Device 4:** Device is currently assigned to a patient with the following description:
*Timesteps wearing the device*: 1
*Pulse rate* - Last value: 80.02 - Mean: 79.03 - Standard deviation/volatility: 1.58
*Respiratory rate* - Last value: 22.71 - Mean: 21.31 - Standard deviation/volatility: 5.45
*SPO2* - Last value: 99.61 - Mean: 99.86 - Standard deviation/volatility: 0.14

**Explanation A:** I chose to reallocate device 4 because the patient currently using it has a stable pulse rate (80.02) and a high SPO2 level (99.61), indicating they are less at risk and may not require continuous monitoring. In contrast, reallocating this device allows for an incoming patient, who likely has more urgent health needs, to receive the device, thereby optimizing the overall allocation of resources to those who require immediate attention.

**Explanation B:** I observed that device 3 is currently free, and there is an incoming patient who requires immediate monitoring to prevent potential deterioration in their health. I used a rule stating that all incoming patients must be assigned a device, with priority given to those at risk. I concluded that assigning 'device': 3 to the incoming patient ensures timely and necessary vital sign monitoring, aligning with the prioritization principle.

**Q1. Do Explanation A and Explanation B appear the same or different to you?**

☐ Same (Skip Q2 and go to Question Q3)
☐ Different

**Q2. Which explanation do you find better?**

☐ Explanation A
☐ Explanation B

**Q3. Do the explanations contain any hallucinations?**

☐ Both
☐ Only Explanation A
☐ Only Explanation B
☐ None