# OpenReview forum: "Rule-Bottleneck RL: Learning to Decide and Explain for Sequential Resource Allocation via LLM Agents"
_ICLR.cc/2026/Conference — ICLR 2026 Conference Withdrawn Submission_

### Official Review · Reviewer_eNFN · 2025-10-18

**Soundness:** 3
**Presentation:** 3
**Contribution:** 2
**Rating:** 6
**Confidence:** 2

**Summary:**

This paper proposes RBRL, which is a LLM agent framework for resource allocation problem that jointly optimizes language-based decision policy and explainability. The LLMs are used to generate a diverse set of rules according to the environment state. The author extend the conventional environmental state-action space by integrating the rules into state generated by LLMs, which make a novel framework that enables RL to operate on a more interpretable decision strucrture. Experiments show the satisfactory performance of the proposed model.

**Strengths:**

1. The proposed model integrates rule generation and selection into the reinforcement learning framework, forming an extended MDP. It employs an attention mechanism to handle dynamically changing rule sets and introduces a "rule reward" to jointly optimize decision-making performance and explanation quality. The idea is interesting and the motivation is novel.
2. The experimental setup is comprehensive, covering three resource-constrained allocation domains such as wearable device assignment, heat alerts, and bin packing. The results are thoroughly analyzed, demonstrating the model's advantages.

**Weaknesses:**

1. The appendix contains material that belongs in the main text, which might violate submission guidelines? Additionally, the paper lacks a conclusion section.
2. The paper does not address the model's computational efficiency, despite each decision requiring multiple LLM calls for rule generation, which may be very time-consuming.
3. The paper does not address how to ensure the LLM-generated rules are both diverse and high-quality. The theoretical assumption regarding rule coverage also appears too strong.

**Questions:**

1. Although the paper emphasizes rules as a "bottleneck", it remains unclear how this bottleneck concretely governs the decision flow. Specifically, is there a strong constraint between the selected rule and the final action? Since the LLM can still freely generate actions after the RL policy selects a rule, this weak coupling may lead to a misalignment between the rule and the executed action, undermining the intended bottleneck effect. Could the authors provide corresponding theoretical analysis on this mechanism?
2. The authors do not systematically evaluate the quality of LLM-generated rules—such as their coverage of optimal decisions, handling of redundancy or conflicts, and temporal consistency. Furthermore, the potential biases of using LLMs as judges remain unaddressed, warranting further human alignment.
3. The computational efficiency of the proposed method remains unexamined. Since each decision step requires multiple LLM invocations, this is likely to lead to significant computational overhead. A time efficiency analysis should be provided.
4. An ablation study is needed to compare the effect of including versus excluding the dynamically generated rule set in the state representation, as this inclusion may introduce noise and hinder policy convergence.
5. Is it over-designed to use ”cross-attention + self-attention” for selecting rules? Could the author compare some simple rule selection strategies?

---

> ### Author Response · Authors · 2025-11-24
> **Response to Reviewer eNFN #1**
>
> Thank you for the constructive feedback and for recognizing the soundness and novelty of our framework. We address your specific concerns regarding the bottleneck mechanism, rule quality, and computational efficiency below.
>
> **R1W1:** We acknowledge the reviewer's concern regarding the structure. Due to strict page constraints, we moved the Impact Statement and Limitations (Appendix C) 1 and the conclusion to the supplementary material. In the final camera-ready version, we will integrate these sections back into the main text to ensure a complete and self-contained discussion.
>
> **R2W2:** We explicitly addressed the computational efficiency in Appendix F.42. Unlike the FinetunePPO baseline which required an NVIDIA A100 40GB GPU, RBRL trains on a standard laptop (using CPU for the RL component). A full training cycle (2 million environment steps) costs less than $10 USD in API credits.
>
> We achieve approximately  million steps in 4 hours using parallelized environments. This is significantly faster than the token-level fine-tuning baseline, which required 18 hours on an A100 GPU for the same number of steps.
>
> **R3W3:** We empirically validated the "Rule Coverage" assumption in Appendix G.5 (Figure 8). We found that as the number of generated rules ($N$) increases (e.g., to 10–20), the probability of the rule set containing a rule that yields the optimal action (derived from an Oracle policy) approaches 100% across all domains. This confirms that the LLM's semantic priors are sufficient to generate diverse, high-quality candidates. In addition, the Rule Reward (Section 4.3) explicitly scores rules based on their predictiveness and applicability. The RL policy maximizes this reward, effectively learning to filter out low-quality rules over time.
>
> **R1Q1:** We explicitly tested this weak coupling in Appendix G.3 by measuring "Action Consistency". We compared the actions taken when the LLM is prompted with `(State + Rule + Reasoning)` versus `(State + Rule)` only.  As shown in Table 8, the Agreement Score is >92% across all domains (and 99% in HeatAlerts). This empirically proves that the Rule is the primary causal driver of the action, proving the coupling is strong.
>
> **R2Q2:** We have conducted systematic evaluations regarding both rule coverage and the validity of the LLM judge in the Appendix.
>
> 1. **Coverage of Optimal Decisions:** We empirically evaluated the quality and coverage of the generated rules in Appendix G.5 (Figure 8). By comparing the rule-based decisions against an Oracle Numeric SAC policy, we found that as the number of candidate rules ($N$) increases (e.g., to 10–20), the probability of the rule set containing a rule that matches the optimal action approaches 100% across all domains 1. This systematically confirms that the un-tuned LLM generates rules with sufficient coverage of optimal decisions.
>
> 2. **Quality and Redundancy:** The quality of the selected rules is ensured by the Rule Reward mechanism defined in Section 4.3. This reward explicitly scores rules on sufficiency and applicability . The RL policy is trained to maximize this signal, effectively learning to filter out redundant, conflicting, or low-quality rules over time.
>
> 3. **LLM-as-a-Judge:** To address the concern of bias, we validated the LLM judge against human preferences in Appendix H. As shown in Figure 9 (Human Survey) and Figure 10 (LLM Survey), there is strong alignment between human evaluators and the LLM judge, with both groups overwhelmingly preferring RBRL explanations over the baselines . This validates that the LLM judge serves as a reliable proxy for human alignment in this context.
>
> **R3Q3:** Please refer to **R2W2**.
>
> **R4Q4:** We respectfully clarify that we did not perform an ablation excluding the generated rule set $\mathcal{R}_t$ from the state representation because doing so would make the learning problem ill-posed and prevent the policy from functioning by definition. In our RBRL framework, the "action space" is dynamic: at every timestep, the LLM generates a new, unique set of natural language rules. Because the semantic content of "Rule 1" changes at every step (e.g., "Prioritize high risk" at $t=1$ vs. "Conserve budget" at $t=2$), the policy cannot learn a mapping from state to action index if it cannot "see" the embeddings of the current options. Excluding the rule set from the state would force the agent to select a rule blindly, reducing the decision process to random guessing among unknown options, which would lead to non-convergence.
>
> Regarding the reviewer’s concern about noise hindering convergence, we address this not by excluding the rules, but by employing the Cross-Attention mechanism (Algorithm 2). This architecture allows the policy to weigh the relevance of each rule embedding dynamically against the state, effectively filtering out "noisy" or irrelevant rules while attending to those that represent optimal strategies.

---

> ### Author Response · Authors · 2025-11-24
> **Response to Reviewer eNFN #2**
>
> **R5Q5:** The use of an attention-based architecture is not over-designed but rather structurally necessary for the specific nature of our data, which consists of sets of natural language rules.
>
> 1. **Theoretical Necessity of Cross-Attention (Permutation Invariance):** The input to the policy is a set of $N$ candidate rules $\mathcal{R}_t$. "Simple" strategies, such as concatenating rule embeddings into a large vector and passing them through a Multilayer Perceptron (MLP), suffer from two critical flaws in this context:
>  *First,* MLPs are sensitive to the order of inputs. If the LLM generates the same set of rules but in a different order (e.g., Rule A then Rule B, vs. Rule B then Rule A), an MLP would treat these as different states. Cross-Attention is permutation invariant, treating the input as a set rather than a sequence, which is the correct inductive bias for this task. *Second,* while we fixed $N$ in experiments, the attention mechanism allows the architecture to handle a variable number of candidate rules without retraining or architectural changes, whereas an MLP requires a fixed input dimension.
>
> 2. **Role of Self-Attention:** We included Self-Attention to capture inter-rule relationships. Since the generated rules often contain redundancies or conflicting logic (e.g., Rule 1 is a special case of Rule 2), Self-Attention allows the policy to reason about the relationships between candidates before comparing them to the state, enabling more robust selection than evaluating each rule in isolation (e.g., via simple dot-product similarity).
>
> 3. **Comparison to Simple Baselines:** We did compare against the "Rule-bottleneck (no RL)" baseline. This baseline effectively represents a "simple selection strategy" where the rule is selected based on the LLM's internal probability logits (its native preference) rather than a learned attention mechanism. The significant performance gap (Figure 4) demonstrates that a learned, context-aware selection mechanism (RBRL) is required to identify optimal rules that align with the specific reward function.

---

### Official Review · Reviewer_Yuu3 · 2025-10-31

**Soundness:** 2
**Presentation:** 2
**Contribution:** 3
**Rating:** 4
**Confidence:** 4

**Summary:**

This paper proposes the Rule-Bottleneck Reinforcement Learning (RBRL) framework, which aims to combine the explainability provided by large language models (LLMs) with the reward-maximizing policy updates of reinforcement learning (RL) for sequential decision-making tasks.
The paper claims that:
-  RBRL improves explainability, as it integrates an LLM component to generate interpretable reasoning.
- It is more efficient than fine-tuning LLMs, which can be computationally expensive and time-consuming.
- The approach still involves updating a policy neural network, and the cumulative reward increases steadily as training progresses.

**Strengths:**

1. The paper presents a novel idea of combining LLMs with RL to address resource-constrained allocation problems. This framework enables LLMs to provide natural language explanations for action choices.
2. It offers a detailed and clear description of how to integrate LLM inference—through prompt selection and standardized schema outputs (rule sets)—with a policy-updating network, resulting in a holistic decision-making policy within an MDP framework.
3. The authors conduct experiments on four datasets: two Wearable Device Assignment tasks (Uganda and MIMIC-III), as well as the HeatAlert and BinPacking tasks.
4. To evaluate the explainability of decision processes, rules, and action suggestions, the authors also conducted a survey with 40 participants to assess the quality of the natural language explanations generated by the LLM.

**Weaknesses:**

1. The reviewer is concerned with the **writing and organization** of the paper. For instance, in line 245, *R* should likely be written as \( R^t \) (if not, clarification is needed on what *R* represents). The paper lacks a clear and intuitive illustration of the RBRL framework where Figure 1 is difficult to interpret and poorly aligned with the description in Section 4. Moreover, Algorithm 1 appears too early, before many variables are defined, which disrupts the reading flow. The transition from Section 5 (theoretical guarantees) to Section 6 (experiments) also feels abrupt; after understanding RBRL, the paper would flow more naturally by moving directly into experiments. Due to page constraints, both the theoretical and experimental sections are underdeveloped—one could be prioritized to better highlight the work’s novelty and contributions. *Minor:* line 440 refers to “MIMIC-II” instead of “MIMIC-III.”

2. The **experimental design and results** raise significant concerns. The shaded areas in Figure 4 indicate high uncertainty. Furthermore, aside from the MIMIC dataset, the non-RL baselines (e.g., CoT or LLM-prompted rule bottlenecks) achieve strong cumulative rewards compared to pure RL methods, which perform worse even after extensive training. This raises the concern that the LLM (billions of parameters) might be the dominant factor driving performance, potentially benefiting from **data leakage** or strong priors inherent in the tasks.

3. The reviewer is also concerned about the **complexity of actions and states**. As the combination of possible actions at each step increases, the number of LLM-generated rules will also grow, leading to increased runtime and larger rule-selection network sizes.

4. While incorporating LLMs into the RL framework indeed improves **explainability**, it also introduces substantial computational overhead. Although the authors claim that RBRL is less resource-intensive than fine-tuning LLMs, there is no comparison of the computational cost between a pure RL system and RBRL. Since RBRL involves multiple LLM calls for text and embedding generation in each episode or exploration step, this overhead could be significant.

**Questions:**

1. If the authors were asked to reconsider the writing structure, how might they reorganize it?
2. The reviewer has more background in the healthcare domain and understands that MIMIC-III is an offline dataset. How does such an offline environment contain all possible next states when diverse actions are applied?
3. Could the authors provide an explanation for the uncertainty region shown in Figure 4?
4. This may be an unusual question, but the reviewer hopes there exists a task where the LLM-only model performs poorly, and combining RL with LLM improves performance to a level comparable with pure RL. Is there such a case?
5. If the authors agree that the chosen tasks are relatively simple and have limited action spaces, what types of datasets or tasks would they explore next? Could the authors provide some intuition on how increasing action and state complexity might affect RBRL?
6. Can the authors provide an estimate of how many compute resources or LLM API calls RBRL typically requires compared to a pure RL method, and also compare the runtime needed to perform the same number of update steps?

---

> ### Author Response · Authors · 2025-11-24
> **Response to Reviewer Yuu3 #1**
>
> Thank you for your feedback regarding the organization and experimental concerns. We address each point below, referencing the specific evidence in our manuscript and appendix.
>
> **R1W1:** We appreciate the detailed feedback on the structure. In the revised manuscript,
>
> 1. We will clarify the notation in line 245 to explicitly denote the reward function $R(s, a)$ or the specific return variable to avoid confusion.
>
> 2. We will move Algorithm 1 to Section 4, placing it after the definition of the key components (Rule Generation, Selection, etc.) to improve the reading flow as suggested.
>
> 3. We will redesign Figure 1 to better visually align with the steps described in Section 4.
>
> 4.  We plan to move the "Impact Statement and Limitations" (currently Appendix C) and the "Conclusion" into the main text to better highlight the contributions and constraints within the page limits.
>
> **R2W2:**  We address the concerns regarding variance and the specific contribution of the RL component versus the pre-trained LLM below.
>
> 1. **Explanation of Uncertainty (Figure 4).** Unlike standard RL, RBRL involves sampling natural language rules from the LLM at each step. The stochastic nature of language generation naturally introduces higher variance in the early stages of training. Despite this variance, Figure 4 shows that RBRL (Red line) demonstrates a consistent upward learning curve and converges to a higher mean reward than the baselines across all domains . This indicates that the RL policy successfully stabilizes the agent's behavior over time by filtering out suboptimal rules.
>
> 2. **LLM Dominance vs. RL Contribution.**  To test if the LLM is the dominant factor, we explicitly compared RBRL against the "Rule-bottleneck (no RL)" baseline (Orange line in Figure 4). This baseline represents the LLM acting solely on its priors. In MIMIC-III, the LLM-only baseline stagnates at a reward of -0.8. RBRL improves this to -0.4. In Uganda, RBRL similarly outperforms the LLM-only baseline significantly. This performance gap (the delta between the Red and Orange lines) isolates the specific contribution of our RL framework. While the LLM provides the candidates (priors), the RL policy provides the optimization necessary to align those priors with the specific reward function of the task.
>
> 3. **Data leakage.** As detailed in Appendix F.1.1, MIMIC-III and Uganda are simulators built on learned transition dynamics (multivariate Gaussian distributions), not static datasets. The agent interacts with generated trajectories that do not exist in the LLM's training corpus. BinPacking and HeatAlerts are defined by specific mathematical reward functions (e.g., minimizing waste subject to capacity $B$). Success here requires dynamic adaptation to the current numerical state, which cannot be achieved via memorization or data leakage.
>
> **R3W3:** We respectfully clarify that increasing action complexity does not necessitate a problematic growth in runtime or network size in our framework.
>
> 1. First, the number of generated rules ($N$) is a user-defined hyperparameter (set to 10 in our experiments 1), acting as a high-level abstraction layer that effectively prunes large, complex raw action spaces into a manageable set of strategic heuristics rather than scaling linearly with every possible atomic action.
>
> 2. Furthermore, our rule-selection policy employs a Cross-Attention architecture (Algorithm 2), which is specifically designed for variable-sized inputs; crucially, the number of learnable parameters in this network is independent of the number of candidate rules ($N$). This ensures that the model size remains constant regardless of complexity, while the inference computation scales only linearly with the fixed $N$, making the method robust for domains with large state-action spaces.
>
> **R4W4:** While we acknowledge that RBRL introduces higher latency than pure numeric RL (e.g., PPO) due to the requisite API calls, this overhead is a necessary trade-off for success in these complex domains. As demonstrated in Figure 4, pure RL methods fail to effectively solve these semantically rich tasks despite their speed. Crucially, our detailed efficiency analysis in Appendix F.4 reveals that RBRL is significantly less resource-intensive than the primary alternative of LLM fine-tuning, which necessitates high-end hardware like NVIDIA A100 GPUs and takes roughly 18 hours per seed. In contrast, RBRL can be trained on a standard laptop CPU , with a full training cycle of 2 million steps requiring approximately 4 hours using parallelized environments and incurring less than $10 USD in API credits.

---

> ### Author Response · Authors · 2025-11-24
> **Response to Reviewer Yuu3 #2**
>
> **R1Q1:**  Please refer to **R1W1**.
>
> **R2Q2:**  We clarify that, as detailed in Appendix F.1.1, our experimental setup does not simply replay historical logs from the offline dataset; instead, we constructed a dynamic simulator derived from the MIMIC-III data . In this environment, the transition dynamics of patient vital signs are governed by a multivariate Gaussian distribution learned directly from real patient trajectories. To support diverse actions, the simulator models the effect of assigning a device by probabilistically adjusting vital signs toward the normal range, specifically modeling an empirical 70% chance of improvement for abnormal vitals, before sampling the next state from the learned distribution . This probabilistic generative process allows the environment to produce valid, counterfactual next states for any action taken by the agent, ensuring the policy is not limited to the specific state-action pairs recorded in the historical logs.
>
> **R3Q3:** As noted in the caption of Figure 4, the shaded regions represent the standard error across 3 random seeds. The variance is naturally higher than in deterministic control tasks because the inputs to the RL policy—the candidate rules—are generated stochastically by the LLM at each step. Despite this variance, RBRL (Red line) shows a consistent upward trend and higher convergence compared to baselines across all domains.
>
> **R4Q4:** Yes, such a case exists, specifically in the MIMIC-III domain shown in Figure 4b. In this environment, the LLM-only baseline (Orange line, "Rule-bottleneck no RL") performs poorly, stagnating at an average reward of approximately -0.8. However, our proposed RBRL method (Red line) drastically improves upon this baseline, converging to a reward of approximately -0.3. In this specific scenario, RBRL not only reaches the performance levels of pure RL but significantly outperforms both Numeric PPO (Blue line, ~-0.6) and Numeric SAC (Green line, ~-0.5). This result explicitly demonstrates that the RL optimization step is crucial for successfully adapting the LLM's semantic priors to maximize the specific reward function of the task.
>
> **R5Q5:** We plan to explore domains with high-dimensional state spaces that require semantic reasoning, such as emergency room triage (with complex patient histories) or supply chain logistics (with qualitative constraints like "fragile"). As the raw action space grows, RBRL becomes more valuable. The "Rule" acts as a high-level abstraction. Instead of exploring a million raw actions, the RL policy only needs to select among $N$ (e.g., 10~100) generated high-level rules, effectively pruning the search space. Increasing state complexity puts more load on the LLM's context window, but the RL policy (which sees embeddings) scales efficiently via the attention mechanism (Algorithm 2), which is designed to handle variable-sized semantic inputs.
>
> **R6Q6:** Please refer to **R4W4**.

---

### Official Review · Reviewer_yoTG · 2025-11-01

**Soundness:** 2
**Presentation:** 3
**Contribution:** 2
**Rating:** 4
**Confidence:** 4

**Summary:**

This paper aims to make more explainable RL policies for decision-making environments. For a given state, an LLM is prompted to generated a set of possible rules to guide the choice of action. Then a policy model is trained to select the most appropriate rule, and another LLM call selects the next action given the state and the selected rule. The rule associated with a decision therefore provides an interpretable justification for the action taken.

The paper is well written and easy to follow, the problem tackled is valuable and the method is interesting. However I have some concerns, mostly regarding the interpretability claims.

**Strengths:**

- The overall objective of explainable decision-making system is valuable and well motivated
- Experiments show good results against naive LLM inference, finetuned LLMs and traditional RL agents.
- Well written paper and easy to follow besides a few presentation issues (see below)

**Weaknesses:**

Major:

- W1: Describing the rules as a bottleneck is misleading. The selection of the action $a^{env}_t$ done by the LLM in Eq 2 includes the full state description $p_t$. Therefore there is no guarantee that the LLM is actually using the rule to decide which action to take. Moreover several works have noted the risks associated with using CoT rationales to interpret an LLM’s output (see for example [1-2]).
- W2: For the same reason as the point above, the explanation $l^{expl}$ has no guarantee of being faithful: beyond hallucinations issues, LLMs can produce plausible but spurious explanations [2].
- W3: l.468, the authors state “explanations follow the State → Rule → Action pipeline, where the rule is the verifiable cause of the action”. Again, this is misleading since the action decision is taken from the LLM with both the state and rule in the prompt, so the causal dependency looks more like “State → Rule, State + Rule → Action”. The actual causal mechanism for the LLM’s action decision is more ambiguous than stated here.
- W4: Given the rule selection is done by the trainable policy in the embedding space, the choice of the rule to base the action choice on is not interpretable. The LLM might be able to provide explanations post-hoc about why such a rule was relevant given the state, but the actual rule selection done by the policy is essentially a black-box.
- W5: In the ablation experiments, a relevant baseline would be replacing the RBRL policy by the LLM choosing among the N generated rules, rather than just generating one rule as in `Rule-bottleneck (no RL)`.
- W6: l.464 “[…] whether an explanation is convincing to humans (validated via our survey, Figure 6)” is overstating - Figure 6 shows that RBRL explanations are merely preferred to CoT explanations.
- W7: The paper does not include a discussion of limitations, nor a conclusion.
- W8: The computational load of the method is significantly larger compared to baselines. Even at test-time, the method requires at least 2 LLM queries at each step (one for rules generation, one for action selection given a rule). This is a major increase in computational overhead compared to standard RL agents like PPO, and might limits the applicability of the method in high-frequency settings. The authors are invited to discuss computational costs in the paper.

Minor:

- l.168: the set $\mathcal{L}$ is not defined anywhere.
- l.171 states “explanations l^expl are generated from actions and thoughts”, but Eq 2 show the explanation as generated the env action and the rule, while l.294 shows it as generated from environment action, rule, and thoughts. Which one is it?
- l.427: the text refers to equations (2)-(5) but I believe the authors mean the four equations under Eq 2.

[1] Lanham, Tamera, et al. "Measuring faithfulness in chain-of-thought reasoning." arXiv preprint arXiv:2307.13702 (2023).

[2] Turpin, Miles, et al. "Language models don't always say what they think: Unfaithful explanations in chain-of-thought prompting." Advances in Neural Information Processing Systems 36 (2023): 74952-74965.

**Questions:**

- l.315: what is $\tau_t$ ? I can’t see it defined or explained anywhere.
- How is it guaranteed, given an LLM, that for any action $a^{env}_t$ there exist a rule $a^{rule}_t$ such that $LLM(a^{rule}_t, p_t) = a^{env}_t $ ?

I might have missed it but this seems like an important assumption for the theoretical guarantees in section 5: if the set of actions accessible in the RBRL MDP $\tilde{\mathcal{M}}$ is different from the original MDP $\mathcal{M}$, then the value of the optimal RBRL policy $\pi_{RBRL}$ would not be guaranteed to upper-bound the optimal policy $\pi^*$ under $\mathcal{M}$ in Theorem 5.3.
- l.430: What does “RulesLLMOnly” refer to?

---

> ### Author Response · Authors · 2025-11-24
> **Response to Reviewer yoTG #1**
>
> We thank the reviewer for the thoughtful and detailed review. We appreciate the positive assessment of our writing and the value of the problem. Below, we address the specific concerns regarding the "bottleneck" mechanism, baselines, and computational costs, pointing to empirical evidence in our Appendix that directly refutes these concerns.
>
> **W1: ...rule bottleneck...**
>
> **Response:** We explicitly tested this "weak coupling" hypothesis in Appendix G.3. We evaluated the "Action Consistency" by comparing actions taken when the LLM sees `(State + Rule + Reasoning)` versus `(State + Rule)` only. As shown in Table 8, the agreement score is extremely high (>92% across all domains, and 99% in HeatAlerts. This empirically proves that the rule, and not the hidden reasoning or the state alone, is the primary causal driver of the decision. If the LLM were ignoring the rule, optimizing the rule selection via RL would not yield performance gains. However, Figure 5a shows that RBRL (optimizing rule selection) significantly outperforms the "Rule-bottleneck (no RL)" baseline, proving that changing the rule effectively steers the action.
>
> **W2: ...Faithfulness...**
>
> **Response:**  We addressed this in Appendix G.4 by masking the action and asking an auxiliary LLM to predict the agent's decision based only on the explanation. Figure 7 shows that RBRL explanations achieve significantly higher accuracy and F1 scores (approaching 1.0) in predicting the action compared to CoT. This confirms that the explanation faithfully captures the decision logic.
> Furthermore, our survey in Appendix H (Figure 11) shows that RBRL explanations have significantly lower hallucination rates compared to CoT (e.g., near 0% vs. 42% in Uganda).
>
> **W3: ... Causal Mechanism Ambiguity...**
>
> **Response:** We conceptualize the decision process not as `State -> Rule` and `State + Rule -> Action` in a loose sense, but rather that the Rule acts as the function (policy logic) and the State acts as the input. The LLM serves as the interpreter that executes `Action = Rule(State)`. While the prompt includes the state (which is necessary for the LLM to ground the rule in specific values, e.g., "vital signs are stable"), the Rule is the only component that changes based on the RL policy's optimization. Therefore, the causal flow of the agent's decision is: `RL Policy -> Selects Rule -> LLM applies Rule to State -> Action.`
>
> We empirically verified that the Rule is the functional driver of the action in Appendix G.3. **Action Consistency (Table 8):** We compared the actions generated by `State + Rule` versus `State + Rule + Reasoning`. The Agreement Score is >92% across all domains (and 99% in HeatAlerts). This confirms that the Rule (applied to the state) is sufficient to dictate the action, and the decision is not driven by hidden reasoning or the state alone. **Translation Fidelity (Appendix G.4):**  We also showed in Figure 7 that the explanation derived from the rule is sufficient to predict the action with high accuracy.
>
> If the causal link were ambiguous, i.e., the LLM might ignore the rule and act based on the state alone, the RL optimization of the rule selection would be ineffective. However, our results in Figure 5a show that RBRL (which optimizes the rule) significantly outperforms the "Rule-bottleneck (no RL)" baseline (where the LLM selects the rule based on priors) . This performance gap proves that changing the Rule effectively steers the Action, validating the rule as the verifiable cause.
>
> **W4: ... Interpretability of Rule Selection...**
>
> **Response:** The goal of RBRL is to achieve intrinsic policy interpretability, ensuring that the agent's high-level strategy is transparent and human-readable at every step. the output of the policy is fully transparent: it is a natural language rule from the set $\mathcal{R}_t$. As shown in Figure 13 (Appendix I.2), these outputs are explicit prioritization strategies (e.g., "Prioritize reallocating devices from patients... with stable vital signs"). This allows a human operator to immediately understand what logic governs the agent's current behavior.
>
> In our framework, the selected rule is the explanation for the action. Unlike post-hoc explanation methods (which generate text to justify a black-box action after the fact), RBRL forces the agent to commit to a readable strategy (the rule) before taking action. Therefore, the "interpretation" of the action choice is the rule itself. The justification for why that rule was selected is provided by the RL optimization process: the policy learned that applying this specific semantic logic in this specific state maximizes the long-term expected reward.

---

> ### Author Response · Authors · 2025-11-24
> **Response to Reviewer yoTG #2**
>
> **W5: ...ablation baselines...**
>
> **Response:**  We understand the reviewer’s concern that the performance gain might stem simply from generating $N$ rules (diversity) rather than the RL selection. In our "Rule-bottleneck (no RL)" baseline (Figure 4, Orange line), the LLM samples a rule from its policy $\pi_{LLM}$. This effectively represents the LLM’s "best guess" or "top choice" based on its pre-trained priors.
>
> *Why RL Selection is Superior to LLM Self-Selection?*  If the LLM knew which rule was optimal for the specific environment reward (e.g., minimizing waste in BinPacking), it would have generated that rule as its top choice in the first place (or assigned it high probability). The fact that the "No RL" baseline performs poorly (e.g., reward -0.8 in MIMIC-III) proves that the LLM's internal evaluation of "good rules" is misaligned with the ground-truth reward function. RBRL does not just pick "the most coherent" rule; it uses Reinforcement Learning to pick the rule that empirically leads to higher rewards, even if that rule is counter-intuitive to the LLM’s priors.
>
> **W6: ...convincing...overstated...**
>
> **Response: ** We agree with the reviewer. We will modify the text in the final version to state that RBRL explanations are "statistically preferred" rather than "convincing," aligning with the survey results in Figure 6.
>
>
> **W7: ...Missing limitation and conclusion...**
>
> **Response:** Due to space constraints, the Impact Statement and Limitations were placed in Appendix C , and the limitations of the simulator/ethics are discussed there. We will move these sections to the main text in the camera-ready version as there is one additional page.
>
> **W8: ...Computation load...**
>
> **Response:** While RBRL is indeed heavier than simple PPO, it is significantly lighter and more accessible than token-level LLM fine-tuning. As detailed in Appendix F.4, RBRL training was performed on a standard laptop (using a CPU for the RL component) and incurred a total API cost of less than $10 USD per training cycle (2 million steps) using gpt-4o-mini. In contrast, the FinetunePPO baseline required specialized hardware (NVIDIA A100 40GB GPU) and took approximately 18 hours per seed, whereas RBRL completed the same number of steps in roughly 4 hours via parallelized API calls.
>
> The reviewer rightly notes that this method may be limited in "high-frequency settings" (e.g., millisecond-level robot control). However, our target domains are high-stakes, lower-frequency resource allocation tasks. For example, for healthcare domain, decisions correspond to patient arrival times. In these contexts, the inference latency of two LLM calls (seconds) is negligible compared to the operational value of safety, interpretability, and trust. We believe the trade-off, i.e., sacrificing millisecond speed for human-readable policy rules, is the correct design choice for these specific real-world applications.
>
> **Response to Minor Points:**
>
> 1. You are correct that $\mathcal{L}$ was not explicitly defined in that line. $\mathcal{L}$ represents the space of all possible natural language strings (thoughts, rules, explanations). We will add this definition.
>
> 2. The correct and most comprehensive formulation is the one used in Algorithm 1 (Line 7) and l.294: the explanation is generated conditioned on the action, the selected rule, the prompt (state), and the thoughts. We will standardize Eq 2 and l.171 to match this description in the revision.
>
> 3. Yes, this refers to the unnumbered equations immediately following Eq (2) that define the policy components. We will fix the referencing.
>
> **Response to Q1:**  $\tau_t$ denotes the "thought" or reasoning trace generated by the LLM via CoT prompting before the rules are generated. This is briefly mentioned in the Figure 1 caption ("produce a thought $\tau_t$") and in Algorithm 1 (Line 3), but we will ensure it is formally defined in Section 3.2.
>
> **Response to Q2:*** We address this in Proposition 5.1 (Rule Set Coverage). We do not assume a single rule always exists; rather, we assume that if we sample a sufficiently large set of candidate rules ($N$) from the LLM, the set will contain at least one rule that maps to the optimal action with high probability ($1 - (1-\eta)^N$). We validated this assumption in Appendix G.5 (Figure 8). The results show that as the number of rules ($N$) increases (e.g., to 20), the "Match Ratio" (probability that the rule set covers the Oracle's action) approaches 100% across all domains. This ensures that the RBRL MDP $\tilde{\mathcal{M}}$ sufficiently covers the action space of the original MDP $\mathcal{M}$.
>
> **Response to Q3:** This refers to the "Rule-bottleneck (no RL)" baseline mentioned in the same paragraph. It was an internal variable name that accidentally remained in the text. We will correct it to "Rule-bottleneck (no RL)" to be consistent with Figure 5a.

---

### Official Review · Reviewer_gFYN · 2025-11-01

**Soundness:** 2
**Presentation:** 2
**Contribution:** 2
**Rating:** 4
**Confidence:** 4

**Summary:**

This paper introduces Rule-Bottleneck RL (RBRL), a framework that combines Large Language Models (LLMs) with Reinforcement Learning (RL) for sequential resource allocation. The key idea is to use a frozen LLM to generate candidate decision rules, while a trainable RL policy learns to select the best rule at each step. This approach jointly optimizes for both task performance and the generation of human-readable explanations, bridging the gap between powerful but opaque Deep RL and interpretable but less effective LLM agents.

**Strengths:**

- The core idea of using a "rule bottleneck" to jointly optimize for decision-making and explainability is innovative.

- The method demonstrates competitive or superior performance compared to several baselines (CoT, PPO) in multiple domains.

**Weaknesses:**

While the proposed architecture is conceptually clear, my primary concern is that it appears over-engineered. This complexity may hinder its generalizability to other domains. Furthermore, its heavy reliance on an un-tuned LLM for core components raises significant doubts about the robustness and generalization capability of the RBRL framework.

Please see my specific questions below for details.

**Questions:**

- The proposed RBRL architecture appears heavily over-engineered, relying on multiple, expensive LLM calls per timestep for rule generation, action selection, and explanation.

-  The method requires multiple LLM calls per step. How slow is it? Please provide actual time costs per step or episode to show it's practical.

- The rules are generated by an un-tuned LLM. How can you guarantee that the set of candidate rules contains good, near-optimal options, especially at the start of training?

- The FinetunePPO baseline is not well-described. Please provide key implementation details.

-  In HeatAlerts and BinPacking, SAC seems to perform worse than the initial policy. Why? Did you perform adequate hyperparameter tuning or longer training for SAC in these environments?

---

> ### Author Response · Authors · 2025-11-24
> **Response to Reviewer gFYN**
>
> We thank the reviewer for your assessment and for recognizing the conceptual clarity of our proposed architecture. We appreciate the opportunity to clarify the practical efficiency of our method and address your specific concerns regarding engineering complexity and baselines.
>
> **Q1&Q2: ...Over-engineering and time costs...**
>
> **Response:**   Contrary to the appearance of high cost, RBRL is significantly more cost-effective than the standard alternative of LLM fine-tuning. As detailed in Appendix F.4, a full training cycle (2 million environment steps) costs less than $10 USD in API credits using gpt-4o-mini. Unlike the FinetunePPO baseline, which required an NVIDIA A100 40GB GPU and ~18 hours of training per seed, RBRL trains on a standard laptop CPU (for the RL component) with parallelized API calls.
>  We achieve approximately 2 million environment steps in 4 hours using parallelized environments. In our target domains (e.g., public health resource allocation or heat alerts), decisions are typically made on an hourly or daily basis. Therefore, the inference latency of generating rules is negligible compared to the value of providing interpretable, safe decisions.
>
> **Q3: ...Un-tuned LLM rules...**
>
> **Response:** We address this in Proposition 5.1 (Rule Set Coverage). Our method does not require all generated rules to be optimal; it only requires that the set of $N$ candidate rules contains at least one near-optimal option . The RL policy then learns to select that specific rule.
> In addition, we empirically validated this in Appendix G.5 (Figure 8). We compared the LLM-generated rules against an Oracle Numeric SAC policy. The results show that as the number of candidate rules increases (e.g., $N=10$ to $20$), the probability of the rule set covering the optimal action approaches 100% across all domains . The LLM’s pre-trained semantic priors allow it to generate reasonable candidates immediately, avoiding the "cold start" problem of pure RL.
>
> **Q4: ...FinetunePPO...**
>
> **Response:**  We have provided detailed implementation settings in  Section 6.1 (lines 432-440), Appendix F.3, and Table 5. We used Low-Rank Adaptation (LoRA) on a Llama 3.1 8B model. A value head was trained on the final hidden states, and we utilized KL divergence regularization from a reference model to ensure stability, following standard PPO fine-tuning practices (Ziegler et al., 2019). As shown in Figure 5b, RBRL outperforms this fine-tuning baseline in terms of sample efficiency and stability, while requiring significantly less compute.
>
> **Q5: ...SAC in HeatAlert/Binpacking...**
>
> **Response:**  These domains involve complex constraints and sparse or delayed rewards (e.g., minimizing waste in BinPacking or managing long-term budgets in HeatAlerts). Pure numeric RL (SAC) struggles here because it must learn the physics of the environment from scratch. In contrast, RBRL leverages the semantic priors of the LLM (e.g., "fit items tightly" or "conserve budget"), allowing it to jumpstart performance.
>
> We utilized the robust hyperparameters from the CleanRL library (Huang et al;, 2022), which is a standard benchmark for these algorithms. We performed adequate tuning (tuning entropy coefficients and learning rates), but the performance gap highlights the fundamental advantage of the RBRLover pure numeric RL in semantically rich tasks.

---

### Note · Authors · 2026-01-22

I have read and agree with the venue's withdrawal policy on behalf of myself and my co-authors.